

# The impact of calibrating soil organic carbon model Yasso with multiple datasets

Toni Viskari[1], Janne Pusa[1], Istem Fer[1], Anna Repo[2], Julius Vira[1], and Jari Liski[1]

[1]Finnish Meteorological Institute, Helsinki, 00101, Finland

[2]Natural Resource Center Finland, Helsinki, 00791, Finland

*Correspondence to*: Toni Viskari (toni.viskari@fmi.fi)

**Abstract.** Soil Organic Carbon (SOC) models are important tools in determining global SOC distributions and how carbon stocks are affected by climate change. Their performances are, however, affected by data and methods used to calibrate them. Here we study how the Yasso SOC model performs if calibrated individually or with multiple datasets and how the chosen calibration method affected the parameter estimation. We found that when calibrated with multiple datasets, the model showed a better global performance compared to a single dataset calibration. Furthermore, our results show that more advanced calibration algorithms should be used for SOC models due to the multiple local maximas in the likelihood space.

## 1. Introduction

Soil Organic Carbon (SOC) models are important tools in estimating current global soil carbon stocks and their future development (Manzoni and Porporato, 2009). Soils are the second largest global carbon pool, hence even small changes in this pool impact the global carbon cycle (Peng et al. 2008). However, SOC and SOC changes are difficult and laborious to measure (Mäkipää et al., 2008). They can also vary drastically over space due to differences in litter fall, site and soil type as well as climate (Jandl et al., 2014, Mayer et al., 2020). Hence, to quantify the global SOC stocks and estimate the effects of different drivers, such as changing environmental conditions, on SOC stocks (Sulman et al., 2018, Wiesmeier et al 2019), numerous SOC models have been developed in the past decades (Parton et al., 1996; Cammino-Serrano et al., 2018; Thum et al., 2019).

Three central research challenges have emerged in the development of SOC models. First, the majority of such models still rely on linear equations representing the movement of C within the soil. This approach has been questioned by the arguments that some of the SOC processes such as the microbial influence or response to a large scale environmental change need to be represented by non-linear equations (Zaehle et al., 2014; Liang et al. 2017) or that the state structure of the model affects which kind of data can be used to calibrate it (Tang and Riley, 2020). More complicated SOC models addressing these arguments have been developed, for example Millennial (Abramoff et al, 2018), and modules including additional drivers affecting the C pools have been included in existing SOC models, such as nitrogen (Zaehle and Friend, 2010) and phosphorus (Davies et al, 2016; Goll et al., 2017) cycles. Their implementation is hindered, though, by the second challenge as data are needed to constrain the model parameterization, but individual measurements campaign datasets are often limited in size and lacking in detail of the SOC state (Wutzlerand and Reichstein, 2007; Palosuo et al., 2012).

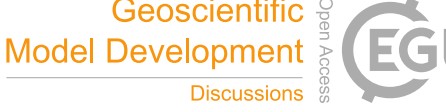

Consequently, multiple datasets should be used to parameterize the models in order to capture the multitude of SOC dynamics, but this naturally raises questions on combining different datasets, and impact of choice of calibration methodology on the resulting parameter sets. Third, decomposition rate dependence on SOC compounds, spatial variation in SOC stocks and different drivers and environmental conditions pose challenges
to SOC model calibration.

While there are measurements that can constrain certain processes, there is no single measurement set that reliably captures all the elements. Litterbag decomposition experiments (Harmon et al., 2009) provide information on the faster decomposition processes, but their applicability to longer-term assessments have been
questioned (Moore et al., 2017). Furthermore, even in current studies it is common to use only data from one litterbag decomposition experiment campaign (for example Kyker-Snowman, 2020) due to the differences in experimental setups and physical properties of the litterbags making direct comparison of results difficult. Organic carbon content can be measured from soil samples, but those measurements provide a limited snapshot because of the large number of measurements needed to detect changes and the slow dynamics of SOC (Mayer
et al. 2020). Additionally, the SOC in these measurements cannot effectively be fractionated into different state components used in the models. Hence, assumptions need to be made on the amount of short-lived SOC to approximate the amount of long-lived SOC. There are also other aspects of litter that are known to affect the decomposition rate, e.g. the bigger the size of the woody litter the slower the decomposition is (Harmon et al., 2000), which requires detailed and specific observations to inform models.

The Yasso07 model (Tuomi et al., 2009) was developed to address some of these challenges. In it, both the litter inputs and the soil carbon are divided into chemically measurable fractions that decompose at their own rate which are affected by environmental conditions, specifically ambient temperature and moisture. This direct link between the model state and litter input allowed using different litter decomposition experiment data to
constrain model parameters. One of the core ideas in the development of Yasso07 is the parameterization process itself is done simultaneously with multiple datasets reflecting different parts of the SOC decomposition process in a Bayesian calibration framework (Zobitz et al., 2011). As a part of this approach, litterbag specific leaching term was introduced in order to be able to use information from several litterbag experiments at the same time (2011b).

While the initial Yasso07 calibration addressed the challenges regarding the variety of data required, it did not touch in detail on the issues affecting the actual SOC model parameterization process. First, the Yasso07 did not calibrate all the parameters simultaneously with all the data, but instead calibrated the parameters in segments where the previously calibrated parameters were set as constant when calibrating the next set of parameters
(Tuomi et al., 2011). While this makes the calibration process easier, it naturally also affects the results and associated uncertainties as well. Second, there has been no standard methods established to evaluate how the inclusion of additional datasets impacts the general performance of SOC models. In other words, does using multiple datasets improve the model estimates? Naturally, this applies to Yasso07 as well. Third, there have been studies which indicate that the choice of parameterization method does matter in ecosystem modelling (Lu
et al., 2017). It is reasonable to assume that would also hold true for SOC systems where there could be multiple





parameter sets that can potentially produce a local fit into the data. Last, but not least, the previous Yasso07 calibration workflow was not easily repeatable and reproducible to allow inclusion of new datasets and algorithms.

In this study, we build upon previous Yasso developments and present a new model formulation and a calibration protocol, that we call Yasso20 hereinafter. Our redesigned calibration protocol leverages BayesianTools R-package (Hartig et al., 2019), an open source general-purpose tool for Bayesian model calibration. Using BayesianTools in our workflow, we not only not only established a more reproducible and standardized application of Yasso20 calibration, but also leveraged interfacing with multiple calibration

algorithms and examined the role of the calibration method.

    Due to the nature of the available SOC related datasets we hypothesize: I) advanced parameter estimation methods account better for multiple potential likelihood maximas than simpler methods, and II) the SOC model performs better globally if multiple datasets are simultaneously used to constrain SOC models compared to a

SOC model calibrated with an individual dataset.

    The first hypothesis is tested by comparing the Yasso parameter values produced by parameter estimation methods of varying complexity and how well they converge. Second hypothesis is tested by calibrating the Yasso with individual datasets as well as the combined data sets with the resulting performances compared

using numerous validation datasets. All these calibrations are done for all the parameters simultaneously. Furthermore, the more extensive calibration process has allowed constraining more details in the new Yasso formulation which is introduced here as well.

## 2. Methods


### 2.1 Yasso model description

    The Yasso model is based on four basic assumptions on litter decomposition and soil carbon cycle: 1) Litter consists of four groups of organic compounds (sugars, celluloses, wax-like compounds and lignin-like

compounds) that decompose at their own rate independent of origin (Berg et al., 1982). 2) Decomposition of any group results either in formation of carbon dioxide ($CO_2$) or another compound group (Oades, 1988). 3) The decomposition rate is affected by environment temperature and moisture (Olson, 1963; Meentemeyer et al., 1978; Liski et al., 2003). 4) The diameter size of woody litter determines the decomposition rate (Swift, 1977). Yasso20 is the next version of Yasso (Liski et al. 2005) and Yasso07 models (Tuomi et al., 2009, 2011b) and

continues to build on these same assumptions. For the purposes of the calibration here, another assumption was necessary: 5) The most stable soil carbon compounds are only formed in the soil as a result of bonding with mineral surfaces (Stevenson, 1982).

    Based on the previously established assumptions, litter can be divided into four fractions according to their chemical composition. Compounds soluble in a polar solvent (water) represent sugars (W) and those soluble in a

non-polar solvent (ethanol or dichloromethane) represent wax-like compounds (E). Compounds hydrolyzable in



acid (for example sulphuric acid) represent celluloses (A) and the non-soluble and non-hydrolyzable residue represents lignin-like compounds (N). Additionally, there is a fifth compartment, humus (H), which represents long-lived, stable soil organic carbon produced by interaction with mineral compounds in the soil. As the carbon compounds are broken down by the decomposition processes, they become either new compounds belonging to

another compartment or $CO_2$. The decomposition rate of each compartment is considered independent of the litter origin and affected by a temperature, moisture, and size component.

The masses ($x$) of the compartments at time $t$ are denoted by vector $x(t) = [x_A(t), x_W(t), x_E(t), x_N(t), x_H(t)]$. The Yasso model uses an annual timestep and determines the changes in those masses according to

$$\frac{\partial x(t)}{\partial t} = M(\theta,c)x(t)^T + b(t), \qquad (1)$$

where $b(t)$ is the litter input to the soil at the time $t$, $\theta$ is the set of parameters driving decomposition as defined in Table 1 and $c$ contains the factors controlling the decomposition. As not only are accurate soil moisture estimates challenging to obtain for the measurements used here, but a vast majority of them are from the surface. Thus, air temperature $T$ and precipitation $P$ were used as the environmental  drivers along with the woody litter diameter $d$. Operator **M** is the product of the decomposition, as presented by K, and mass fluxes

between compartments, as depicted by F, equations as follows

$$M(\theta,c) = F(\theta)K(\theta,c),$$
(2)

$$F(\theta) = \begin{bmatrix} -1 & p_{WA} & p_{EA} & p_{NA} & 0 \\ p_{AW} & -1 & p_{EW} & p_{NW} & 0 \\ p_{AE} & p_{WE} & -1 & p_{NE} & 0 \\ p_{AN} & p_{WN} & p_{EN} & -1 & 0 \\ p_H & p_H & p_H & p_H & -1 \end{bmatrix}, \qquad (3)$$

$$K(\theta,c) = diag, \qquad (4)$$

Here parameters $p_{ij} \in [0,1]$ denote the flows from compartment $i$ ($i \in \{A,W,E,N\}$) to $j$ ($j \in \{A,W,E,N,H\}$) and are included in the parameter vector $\theta$. The decomposition rates $k_i(\theta,c)$ were calculated according to

$$k_i(\theta,c) = \frac{\alpha_i}{J} h(d)(1 - e^{\gamma_i P}) \sum_{j=1}^{J} e^{\beta_{i1} T_j + \beta_{i2} T_j^2}, \qquad (5)$$

where the base decomposition rate $\alpha_i$, temperature parameters $\beta_{i1}, \beta_{i2}$, and precipitation parameter $\gamma_i$ for

compartments $i \in \{A,W,E,N,H\}$ are all a part of the parameter set $\theta$. The temperature and precipitation dependent rate parameters are the same for compartments AWE, but both N and H compartments are given their own separate parameter values. In order to capture the annual temperature cycle more efficiently, the average monthly temperatures for all 12 months are given as an input with the model averaging over their impacts as seen in eq. (5). The total annual precipitation is used instead of monthly precipitation as seasonal variation such

as snowfall or heavy rainfall followed by long dry stretches would hinder the calibration if the monthly precipitation was used. The temperature and precipitation equations are established in Tuomi et al. (2008). Woody litter decomposition rate in response to diameter ($d$) is described in $h(d)$ based on Tuomi et al. (2011), as follows,





$$h(d) = min((1 + \varphi_1 d + \varphi_2 d^2)^r, 1), \tag{6}$$

where $\varphi_1$, $\varphi_2$, and $r$ are parameters included in the parameter set $\theta$.

Given initial state $x_0$, average environmental conditions $c$ and constant litter input $b(t) = b$, the model prediction can be computed by solving the differential equation in Eq. (1). The solution becomes

$$x(t) = M(\theta,c)^{-1}\big(e^{M(\theta,c)t}(M(\theta,c)x_0 + b) - b\big), \tag{7}$$

where the matrix exponential is determined numerically. In a steady state situation $x = \lim_{t\to\infty} x(t)$, equation 7 becomes

$$x = -M(\theta,c)^{-1}b, \tag{8}$$

### 2.1.1 Yasso20 improvements

Two main changes were introduced to the Yasso20 version here compared to the earlier Yasso07 version. The first change was that the temperature input for Yasso20 is given as the mean monthly temperature for each month of the year instead of the mean annual temperature and associated annual temperature amplitude. This was done in order to better represent the more nuanced global temperature profiles. For example, the previous scheme was indifferent if the winter was long or short, which is, however, expected to affect the annual decomposition. The second change was to differentiate the climate driver impacts between the AWE, N and H pools instead of using the same parameter values for all the model C pools. This was done because previous research established that more complex carbon compounds require more energy to be broken up (Davidson and Janssen, 2006), which indicates that the parameters representing those dynamics should also differ between pools. It is expected that these changes will affect the model performance and the calibration results themselves, especially as this allows the environmental conditions to impact the pools differently. Thus this changed model version was decided to be a new version of the model. We do not compare Yasso20 performance to Yasso07 here. All model parameters given in Table 1 were targeted in the calibration.

### 2.2 Datasets used in the calibration

Several datasets were simultaneously used to calibrate the model in order represent different processes related to soil carbon cycling: Decomposition bag time series data from the Canadian Intersite Decomposition Experiment (CIDET; Trofymov, 1998), Long-Term Intersite Decomposition Experiment (LIDET; Gholz et al, 2000) and European Intersite Decomposition Experiment (ED; Berg et al., 1991a, 1991b) projects, a collection of global soil organic carbon measurement gathered by Oak Ridge National Laboratory (Zinke et al., 1986) and woody matter decomposition dataset from Mäkinen et al. (2006). In addition to these large datasets, a smaller litter bag decomposition data set from Hobbie et al. (2005) was used to both evaluate how much addition of a comparatively small number of data points affects the calibration results as well as an independent validation dataset for the other calibration parameters. These datasets along with additional details are listed in Table 2.





CIDET, LIDET and ED are litter bag decomposition timeseries where litter is left to decompose in a mesh bag
        and the remaining mass is measured at chosen time intervals over several years. Each dataset had the
        experiments with multiple different species, with the initial chemical composition also provided by the dataset,
        and different sites. Furthermore, while CIDET and LIDET only measured the remaining mass, ED also
        determines the AWEN fraction from one of the replicant samples, which allows us to directly compare it to the
Yasso20 state variables. However, while in CIDET and LIDET the remaining mass has ash removed, in ED ash
        is still included in the remaining mass. The mean monthly temperatures and precipitations have been measured
        at each test site with the annual precipitation being summed up from the monthly precipitation values.

        The global SOC measurement dataset from Oak Ridge National Laboratory (Zinke et al., 1986) is collected
from the data of numerous unrelated projects that have measured SOC as a part of their campaign. As such,
        there are/were no uniform applicable protocols to these measurements. For the purposes of the calibration, the
        data is assumed to represent the steady state SOC at that location and each measurement is treated as
        independent from the others even if they are from the same location. Furthermore, we only used SOC
        measurements that were below 20 kgC m$^{-2}$ in the calibration. Values higher than those were found in high
latitudes and considered as results of waterlogging, peat formation or permafrost, processes not described in
        Yasso20. The litter input was determined by combining the global GPP map from Beer et al. (2010) with the
        global NPP/GPP relationship set to 0.5 at the measurement locations due to lack of specific information on the
        NPP/GPP there. The Olson classification (Olson et al., 2001) regarding the local ecosystem type was used to
        roughly divide the ecosystems into grasslands, semi-forests and forests. The litter fractioning for these different
systems are given in Supplemental Table 1. In addition, SOC chronosequence data from Liski et al. (1998) and
        plot level measurements of Liski and Westman (1995) was used as a validation data set.

        The woody decomposition data used here is from Mäkinen et al. (2006), which has measurements of multiple
        trees in different stages of decomposition over several decades in Finland. There are no signifiers to connect the
measurements from different years nor to indicate how much the tree diameter has been reduced over time
        because the data was not chronosequence data of the same trees. As such, the measurements were considered
        independent and representative of decomposition of a tree trunk of that size.

### 2.2.1 Dataset uncertainties


        The information of the uncertainty related to the measurements was limited. With CIDET and LIDET there are
        generally four replicants, sometimes less, from which the standard deviation in remaining mass can be
        calculated. Similar standard deviation is available for the ED measurements, but is only determined for the total
        mass loss and not for the AWEN pool measurements used here. Furthermore, there are other aspects affecting
the uncertainties such as the ED measurements containing ash or LIDET measurement time series showing more
        noise than the CIDET measurements. For the global SOC dataset and the woody matter decomposition datasets
        no such replicant deviation is available nor is there any other established uncertainty. There are other similar
        measurement campaigns where uncertainty estimates are given, but it is not clear how directly they can be
        applied for the datasets used here. Consequently, here we used our expert opinion to determine the different



dataset uncertainties relative to each other (Table 1) as we felt this was a more transparent manner to
acknowledge the current limitations regarding assigning the uncertainties.

Systematic differences in the litter bag properties affected the use of different datasets (Tuomi 2009;
Tuomi2011b). In general, high mass loss rates were positively correlated with a large mesh size of the litter bags

and high precipitation in our datasets. This is because the decomposing material in the litter bags is partially
'washed away' into the surrounding soil by water flow and is thus removed from the bag due to processes other
than decomposing. To correct for this, we added a leaching term to equation 1 as follows,

$$\frac{dx(t)}{dt} = (A(\theta, c) - \omega_{site}PI_5)x(t) + b(t), \tag{9}$$

where $\omega_{site}$ is the dataset-specific leaching term and $\mathbf{I}_5$ is a 5×5 identity matrix. This approach was simplified as

there are multiple components expected to affect the leaching process and other systematic errors, but it was
necessary to establish even this simplistic initial approach for the work here.

Finally, long-lived carbon compounds represented by the H pool in the Yasso model are not produced in
decomposition litter bags as they require organo-mineral associations which are unlikely to occur in the litter

layer"

that is only possible in the soil. Because of this $p_H$ (transfer fraction from AWEN pools to pool H) could have
non-zero values only with the Oak Ridge global SOC dataset.

**2.3 Calibration protocol**

We used the BayesianTools R-Package (Hartig et al., 2019) in our calibration workflow for its standardized and
flexible implementation of Markov chain Monte Carlo (MCMC) algorithms with external models, as well as for
its post-MCMC diagnostic functionality. While our main aim in this paper was not to compare MCMC
algorithms, once the interface was established with the BayesianTools, it was trivial to leverage the common

setup and test the performances of different MCMC flavors as implemented by the package. We found this
exercise helpful as our calibration problem involves a relatively high dimensional and irregular likelihood
surface. It has been previously shown that for such systems the efficacy of the calibration may differ between
algorithms (Lu et al., 2017). Thus, we tested two robust and efficient algorithms Differential Evolution Markov
Chain with snooker updater (DEzs, ter Braak and Vrugt, 2008) and Differential Evolution Adaptive Metropolis

algorithm with snooker updater (DREAMzs, Vrugt et al., 2009; Laloy and Vrugt, 2012; Vrugt, 2016), in
addition to the long-established adaptive Metropolis (AM) algorithm (Haario et al., 2001).

All three algorithms use Markov chains to explore the parameter space and generate samples from the posterior.
However, AM uses a single chain, whereas DEzs and DREAMzs use multiple interacting chains simultaneously.
While DREAM emerged from DE, DREAM further uses adaptive subspace sampling to accelerate convergence

(Vrugt, 2016). All three algorithms use proposal distributions to generate successive candidate samples and
grow the chains. However, AM uses a multivariate Gaussian distribution as the proposal which is most effective

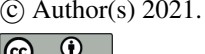



when the target distribution (a.k.a. posterior) is also Gaussian. Whereas, DEzs and DREAMzs algorithms use the differential evolution principle to optimize the multivariate proposals (with snooker jumps to increase the diversity of the proposals), automatically adjust the scale and orientation of the proposal distribution according

to the target distribution (Vrugt et al., 2009; 2016). As a result of these properties, especially when not tuned properly, AM can take much longer to complete the high-dimensional parameter search and can suffer from premature convergence when multiple distant local optima are present (Vrugt, 2016; Lu et al., 2017). Whereas DEzs and DREAMzs can potentially resolve non-gaussian, high-dimensional and multimodal target distributions more effectively without much configuration (Laloy and Vrugt, 2012, Lu et al., 2017).


In our calibration protocol, we ran 3 chains for each algorithm where DEzs and DREAMzs further tripled each chain. We initialized these chains from the prior distributions (Table 1) using the random sample generator of the BayesianTools package. Each chain was run for $1.5 \times 10^6$ iterations and the last $1.5 \times 10^5$ iterations were used to compute the posterior probability distributions after removing the burnin. Convergence diagnostics were

checked by visually inspecting the trace plots of the chains, as well as calculating the multivariate R-statistic of Gelman and Rubin (1992).

Initially the calibration was done with all the parameters associated with the Yasso20 model. However, if the estimated parameter values for the p-terms in eq. 3 were within three decimals from either 0 or 1, they were set

to nearest limit value of 0 or 1, after which the calibration was redone. During the calibration, the p value parameterization can never settle at 0 or 1 and, hence, it is impossible to know what the real p value is that close to the limit. The calibration results presented here only had four p values that were not set: $p_{WA}$, $p_{WN}$, $p_{EW}$ and $p_{EA}$. Parameters $p_{AW}$ and $p_{NA}$ were set to 1 and the other AWEN related p values were set to 0. Furthermore, since we assumed that only decomposition in the W pool results in CO2, we estimated only $p_{EW}$ and set $p_{EA}$ to

be the E-pool remnant from 1 with $p_{EN}$ set to 0.

### 2.4 Validation protocol

Each of the litter decomposition experiments (CIDET, LIDET and ED) was randomly split into two: data used for calibration (80% of the measurements) and data used for validation (20% of the measurements).

Furthermore, the random division is done so that the whole measurement time series from one bag is always fully either in calibration or validation data. It was also verified that each site and species in was approximately equally represented in both the calibration and the validation data. Due to the noise and bias in both the global SOC measurement data sets in addition to the separate processes included in those calibrations, we did not divide them into calibration/validation parts but used all the data for calibration.

The experiments were conducted by calibrating the Yasso model individually with the calibration data from each litter bag decomposition data set (CIDET-only, LIDET-only, ED-only) as well as a joint calibration that used all the calibration data detailed before (i.e. CIDET, LIDET, ED, Mäkinen, global SOC). The leaching parameter was individually calibrated for each decomposition bag dataset during the joint calibration. In addition, the Hobbie3 dataset (Hobbie, 2005) was used as an independent validation dataset. Since there was no





information on its leaching parameter, that was set to zero in the validation runs. The validation for each
       calibration results was done with all the separate validation data sets. Similar validation dataset is created with
       the Mäkinen wood decomposition data with 20 % of the data points set aside for validation purposes. There was,
       however, no independent calibration done with the Mäkinen dataset as there is not enough of data there to
       constrain the model completely and in the validation analysis the focus was on how it performed over wood size
instead of time.

       The global Oak Ridge SOC data set was not split into calibration/validation parts for two reasons. First, as it
       was the only dataset calibrating the H parameters, there was no efficient to way to evaluate how the addition of
       new data would have impacted the model performance regarding this dataset. Second, the dataset was found to
       be so noisy that the randomized choosing of the validation datapoints already affected the results to a noticeable
degree. Due to this, the H parameter calibration was evaluated with two separate small datasets. First, SOC
       measurements from several plots Hyytiälä, Finland (Liski & Westman, 1995) where the dominant tree species of
       each plot is known was used to test if Yasso20 was able to calculate an approximately correct SOC value for the
       plots. The SOC values for plots with the same dominant species were averaged for the comparison with the
       litterfall used for each species listed in Supplemental Table 2. Second, a SOC chronosequence from Liski (1998)
was used to determine if Yasso20 is able to realistically simulate the SOC accumulation over time scales of
       hundreds of years. In this dataset there are 26 soil age gradient data points from the Finnish coast which has
       been used to approximate the SOC accumulation in the soil over hundreds of years after the ice age. Tree litter
       and climate driver data from Hyytiälä, Finland was used here as the main focus is on if the simulated system
       reaches steady state in the same time window as the measurements. The climate driver data used for these
validation runs is included in Supplemental Table 3.

## 3. Results

### 3.1 Calibration

All three calibration methods (AM, DEzs, DREAMzs) produced similar maximum a posteriori (MAP) values
       for global (joint) calibration where all data streams were used (Table 3). Closer examination of different chains
       (Figure 1), though, shows that while DEzs and DREAMzs converged to the same parameters, AM chains
       instead produced different parameter distributions. The Gelman-Rubin (G-R) statistics for the different
       calibration methods (Table 3) reflect these differences in convergence as well, with DEzs having the values
within the acceptable boundary while values for AM are above acceptable ranges. DREAMzs also performs
       generally well but shows more divergence with the parameter values than DEzs. Similar behaviour was seen
       when running the individual dataset calibrations, where individual AM chains would mix well, but converged at
       different values from each other (Not shown). Per global calibration diagnostics of different algorithms, we
       decided to report the rest of the results with the DEzs algorithm for clarity as its estimates were converging best
out of the three examined methods. When the global calibration with the DEzs algorithm was repeated with the
       Hobbie3 data set included, the resulting parameter distributions were nearly identical to the calibration done
       without the Hobbie3 data set included (Not shown).



When comparing calibrations done with the DEzs approach for the individual data sets as well as the global
        calibration that used all that data (Figure 2; Table 4), the parameter sets produced by the calibrations differ from
        each other to a meaningful degree. For each dataset, the RMSE values are at their lowest when using the
        parameter sets calibrated with that specific dataset (Table 5), though the global parameter set produced RMSE
        values close to those lowest values. However, when using the parameter sets calibrated by other datasets than
        the validation data have been chosen from, the RMSE values became higher indicating worse model

performance. When the RMSE analysis was done with the Hobbie3 dataset, the global parameter showed the
        best performance. It should be noted that since in the ED data set measurements are for each individual AWEN
        pool, the individual measurements are smaller in value than the total mass measurements of
        CIDET/LIDET/HOB3. Consequently, the RMSE values for ED are smaller than those for
        CIDET/LIDET/HOB3 datasets.


        With regard to the long-term SOC projections, the comparisons with the Hyytiälä forest plot measurements
        (Table 6; Figure 3) indicates that at least in the Nordic forests Yasso20 potentially slightly overestimates the
        steady state SOC, with the largest differences still being below 2 kg C m$^{-2}$. It should be noted, though, that there
        is notable variance within the measurements in addition to the uncertainty related to the driver data. The

chronosequence data (Figure 4) shows that the model projection saturates approximately in 1000 years similarly
        to the measurements.

### 3.2 Residual analysis

When checking residuals from the litter bag experiments against mean annual temperature, annual temperature
        variation and total annual precipitation (Figure 5), there appears to be a tendency for Yasso20 to increasingly
        underestimate the remaining litter bag C with growing average mean temperature and precipitation. The error
        does not, though, show any signal when looking at the temperature variation within the year. With the woody
        decomposition residuals (Figure 6), there is a slight negative trend over time and a slight positive trend over

size. Both are minor, though, and the residuals for the woody decomposition are relatively evenly distributed for
        the validation dataset.

### 3.3 Parameter values correlation

Analyzing the correlations between different parameter values produced by the DEzs algorithm (Figure 7)
        shows that the correlations are the strongest between processes affecting the same pools. The p-terms which had
        been set to 0 and 1 were excluded from the correlation analysis since they did not vary during the calibration.
        The AWE pools decomposition rates have strong positive correlations between the decomposition rates as well
        as with the climate driver terms affecting decomposition in them. Similarly, there are strong negative

correlations between the temperature terms affecting the same pools and a strong positive correlation between
        the H pool terms. There are both strong positive and negative correlations with the size related parameters.





## 4. Discussion


### Calibration method

Here we showed that by using a DEzs calibration algorithm, we were able to simultaneously use multiple different types of datasets to constrain the soil organic carbon (SOC) model Yasso and produce a converging

parameter set. Additionally, using a more conventional model calibration approach, here the Adaptive Metropolis (AM), showed that it was vulnerable to the local likelihood maximas and that the resulting parameter sets were strongly affected by the starting values. This supports our first hypothesis that more advanced calibration methods should be used to estimate SOC model parameters due to the numerous local likelihood maximas, especially when detailed algorithm configurations are not always possible or desirable. Furthermore,

even the more stable calibration method produced different results for different individual datasets used to calibrate. The global calibration set, in turn, proved to be most efficient across different datasets, indicating that multiple datasets should be used to constrain SOC models. More advanced calibration methods, though, then need to be applied to minimize the impact of the resulting uneven parameter space and producing Gelman-Rubin values within more acceptable ranges (Gelman and Rubin, 1992). Something that was curious in our

results was that DEzs converged better than DREAMzs (Table 3) despite the latter being a more state-of-the-art method (Vrugt, 2016). We were not able to determine the reason for this in our tests here, specifically was it something related to the behaviour of the parameter space or to how the method is implemented in BayesianTools.

### Impact of prior parameter information

One of the fundamental challenges for calibrating SOC models is lack of experimental information regarding the model parameter value distributions. Therefore, we used generally broad uniform prior distributions for the calibration here. However, it is still important to evaluate the calibration results based on our understanding of

the overall system behaviour. For example, initially we used wider priors for parameters $p_H$ and $\alpha_H$ (Results not shown), which in turn resulted in the calibration producing a $p_H$ value of ~0.08 and, consequently, a much higher H pool decomposition rate. As this did not fit with the system behavior seen i.e. with the bare fallow experiments (Menichetti et al., 2019) or the soil chronosequence (Fig 3-3), we applied a narrower prior constraint on the related parameters. Another, and a more, complicated example is that when using wider prior

constraints for the N pool decomposition rate parameter $\alpha_N$, the calibration resulted in the N pool being largely insensitive to the temperature and moisture drivers. While there are no direct measurements of the lignin pool temperature sensitivity, there have been studies showing that the energy needed for breaking down SOC compounds increased with complexity (Davidson and Janssen, 2006; Karhu et al., 2010) indicating that the N pool should be temperature sensitive. Here we chose to constrain $\alpha_N$ to a lower range, which in turn forced a

climate driver sensitivity for it. All these examples illustrate that the calibration results themselves should further be reassessed in their physical meaning.

### The benefit of calibrating with multiple datasets and further inclusion of smaller datasets





Our results show that simultaneously using multiple datasets from different environments improves the general applicability of the SOC model confirming our second hypothesis. This is in line with prior studies arguing for larger representation in the calibration data (Zhang et al., 2020). However, even with this global calibration, individual locations can be affected by specific SOC decomposition conditions not currently accounted for in the models (Malhotra et al., 2019). Naturally, if smaller datasets of SOC and decomposition measurements are

available from locations affected by specific decomposition dynamics, for example agricultural soils that are treated in a very specific manner, it would be logical to use that local information to constrain the SOC model to better suit that location. However, the results here raise questions on how those smaller datasets should be implemented in the model calibration. The inclusion of the Hobbie3 dataset did not meaningfully impact the calibration results (Not shown), which is reasonable considering how small that dataset (N=192) is compared to

the totality of the other datasets (N=~17 000) being used in the calibration. This indicates that due to the sheer size of the global calibration data set, smaller local data sets cannot effectively be used just by adding it to the joint calibration process.  There are other options, though, by either using the globally estimated parameter ranges as the priors for a calibration with the local data or employing a hierarchical calibration approach (Tian et al., 2020, Fer et al., 2021), but the impact of these approaches should be separately researched and tested. Our

study still successfully provided a global parameter set that increases the applicability of Yasso model and informs global SOC estimates.

**Leaching**

As established in section 2.2, in order to compare the measurements from different litter bag experiments, there needs to be a parameter that accounts for the litter bag types' impact on the mass loss rate (Tuomi et al., 2009). When testing with independent litterbag data, we see that even with this added assumption, the global calibration produces a better fit than the calibration based on individual litterbag campaigns (Table 5). This supports using data from multiple litterbag campaigns in model calibration However, in the results it is evident

that not only are the leaching parameters estimated to be essentially zero when calibrating only with individual decomposition bag data sets (Table 4), but also when simultaneously calibrating with all the data sets, only the ED dataset ends up having a meaningfully non-zero value. First of all, this indicates the current straight-forward formulation for leaching is insufficient as with the individual dataset calibrations the other parameter values are able to produce fits where there is no leaching despite knowledge that it is a factor. Second, even when

calibrating multiple data sets simultaneously, the calibration appears to  apply the leaching effect to only one of the datasets even when it should affect all of them. Third, the further complication is that by being a corrective term, the leaching term will also reflect all other systematic differences between the datasets. As a consequence of all these factors, the leaching impact needs to be further studied and the relevant equations need to first be formulated with experimental data specifically gathered for that purpose.


**Humus formation and the need for the layer Yasso**



There is an important point concerning the parameterized humus (H) formation term $p_H$ here. The long-term H formation can only take place in the soil itself as it requires the presence of mineral compounds (Schmidt et al.,

2011), which is why only the global soil carbon dataset in this study could be used to constrain H parameters. However, they are only point measurements with no information of how the state changes over time. Therefore, we have to assume that the measurements represent the approximated steady state from an assumed litter fall. This not only causes larger parameter uncertainties, but also the estimated pH parameter value will represent the fraction of the total litter fall that ends up in the H pool while in reality with the surface vegetation litter there

needs to be an additional mechanism that transfer the carbon compounds to soil while root litter is already in that environment. Consequently, if examining litter decomposition taking place only in the soil, such as with roots, it is likely that pH for that soil system would be larger than what is estimated here. This would fit with previous research suggesting that the root biomass specifically appears to be connected to the amount of long-term carbon in the soil as more of it would be able to form H compounds than the surface vegetation

(Clemmensen et al., 2013; Jackson et al., 2017). However, currently the amount of data that would allow efficiently separating the above and below soil decomposition processes during the calibration process is limited. Additionally beyond this, there are presence of mineral compounds and other conditions that affect how efficiently H is formed that should be included when formalizing H formation (Rasmussen et al., 2018). Better addressing the formation of H is a crucial development step for the model, but the current approach provides an

initial way to estimate the H pool size/quantity.

**Temperature and precipitation impact**

At first glance it appears that the current version of Yasso20 overestimates SOC decomposition (i.e.

underestimates SOC amount) at higher precipitation and temperature values, as indicated by the negative trend in Fig 3-4. In the current formulation of environmental drivers (eq. 5), only the lower precipitation values decrease the decomposition rate with the system becoming insensitive to increases in precipitation after a certain threshold. However, it is known that at higher moisture levels the SOC decomposition rates decrease (Keiluweit et al., 2017). A more informative driver of moisture conditions (e.g. monthly soil moisture) and a more realistic

response function could help disentangle the reasons behind this trend in the residuals in the future. The current version of Yasso20 uses precipitation as the driver instead of soil moisture because the decomposition bags from the data sets used as constraints here are on the surface and thus were expected to be primarily controlled by precipitation. In the light of current findings, next steps in Yasso model development towards using soil moisture as model drivers are planned.


Closer examination of the error distribution over the climate drivers, though, suggests some more complexity. Even at the lower precipitation values while both CIDET and ED data errors cluster approximately equally around zero, the LIDET data points show a shift towards negative errors similarly to at higher precipitation values. Thus, it appears that the issue is at least partially due to the data set itself rather than the pure

precipitation signal. Similar behaviour can be seen with mean temperature, although it isn't as pronounced. Thus, there is a seeming systematic error when simulating the LIDET data with the global calibration parameter





sets. It is yet unclear if this is due to something with the measurements, something with the processes or if the climate driver data is not similarly representative of the conditions as with the other used data sets.

**Litter size impact on decomposition**

In the current Yasso20 implementation, the woody litter diameter does not change during the decomposition process while in reality the wood shrinks as it decomposes. This explains why when comparing the model results to the tree decomposition validation dataset (Fig 3-5), the model overestimates the decomposition rate for
decades old tree stems with a measured diameter of approximately 10 cm. In those cases, the model assumes that was the size of the trunks when the decomposition started and, consequently, the size impact is smaller than it should be. While the model still performs well with the validation database regardless of this, it is an important aspect to consider when applying Yasso20 model with woody decomposition.


**5. Conclusions**

Soil organic carbon (SOC) models should be constrained by data from multiple different ecosystems and reflecting the various dynamics affecting the SOC decomposition process. Using data from multiple datasets
produced parameter sets which performed better in a global comparison than parameter sets calibrated with information from individual datasets, highlighting the necessity of using more data. However, the traditional AM calibration method had difficulties converging to a single parameter set when used with multiple datasets, most likely due to the numerous local likelihood maximas within the likelihood space, and our deliberate choice for avoiding detailed algorithm-specific configurations which reduces repeatability and re-applicability.
Consequently, our results showed that more advanced methods such as DEzs should be used when calibrating SOC models. Furthermore, we identified numerous aspects where further detailed data is needed to better constrain the model processes in question, for example regarding the leaching parameter that allows comparison of different litter decomposition bag experiments or better connecting varying soil moisture conditions to changes in SOC.


**Acknowledgments**

This research has been funded by The Strategic Research Council at the Academy of Finland [decision no 327214 and 327350] and the Academy of Finland Flagship Program [decision no 337552]. This study was additionally supported by CO-CARBON project funded by the Strategic Research Council at the Academy of
Finland [grant no 335204]. A.R has also been supported by the grant [Trade-offs and synergies in land-based climate change mitigation and biodiversity conservation decision 322066 by the Academy of Finland.]

**Author contributions**

Dr. Toni Viskari came up with the study, planned the experiments and wrote majority of the manuscript. M.ScT. Janne Pusa did the simulations and the initial analysis of the results. Dr. Istem Fer participated in the experiment
setup, analysis as well as wrote a part of the methodology. Drs. Anna Repo and Julius Vira provided insight to



the data used and the mathematical aspects of the methods, respectively. Prof. Jari Liski is the PI of the project this research was a part of and has created the Yasso model used here.

**Data availability**

The Yasso model used here can be downloaded from https://github.com/YASSOmodel/Ryassofortran. The permanent version of the Yasso code, data used in this publication as well as the calibration algorithm has also been uploaded to Zenodo (10.5281/zenodo.5059909). The manual on the calibration process is within the calibration folder.

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



| Parameter symbol | Parameter description | Prior distributions | Starting values |
|---|---|---|---|
| $\alpha_A$ | Base decomposition rate for pool A (1/year) | U(0,2) | 1.86, 0.23, 1.37 |
| $\alpha_W$ | Base decomposition rate for pool W (1/year) | U(0,10) | 3.52, 6.0, 9.74 |
| $\alpha_E$ | Base decomposition rate for pool E (1/year) | U(0,2) | 0.36, 1.63, 0.82 |
| $\alpha_N$ | Base decomposition rate for pool N (1/year) | U(0,0.1) | 0.01, 0.06, 0.03 |
| $\alpha_H$ | Base decomposition rate for pool H (1/year) | U(0.001,0.01) | 0.0024, 0.0094, 0.0045 |
| $p_{AW}$ | Transference fraction from pool A to pool W | U(0,1) | Set value of 1-$p_H$ |
| $p_{AE}$ | Transference fraction from pool A to pool E | U(0,1) | Set value of 0 |
| $p_{AN}$ | Transference fraction from pool A to pool N | U(0,1) | Set value of 0 |
| $p_{WA}$ | Transference fraction from pool W to pool A | U(0,1) | 0.31, 0.37, 0.68 |
| $p_{WE}$ | Transference fraction from pool W to pool E | U(0,1) | Set value of0 |
| $p_{WN}$ | Transference fraction from pool W to pool N | U(0,1) | 0.42, 0.45, 0.20 |
| $p_{EA}$ | Transference fraction from pool E to pool A | U(0,1) | Set value of 1-$p_{EW}$-$p_H$ |
| $p_{EW}$ | Transference fraction from pool E to pool W | U(0,1) | 0.47, 0.91, 0.04 |
| $p_{EN}$ | Transference fraction from pool E to pool N | U(0,1) | Set value 0. |
| $p_{NA}$ | Transference fraction from pool N to pool A | U(0,1) | Set value of 1-$p_H$ |
| $p_{NW}$ | Transference fraction from pool N to pool W | U(0,1) | Set value of 0 |
| $p_{NE}$ | Transference fraction from pool N to pool E | U(0,1) | Set value of 0 |
| $p_H$ | Transference fraction from AWEN pools to pool H | U(0.001,0.01) | 0.0071, 0.0064, 0.0026 |
| $\beta_1$ | The first order temperature parameter for AWE pools (1/C) | U(0,0.2) | 0.03, 0.04, 0.17 |
| $\beta_2$ | The second order temperature parameter for AWE pools (1/C$^2$) | U(-0.05,0) | -0.013, -0.007, -0.003 |
| $\beta_{N1}$ | The first order temperature parameter for N pool (1/C) | U(0,0.2) | 0.12, 0.01, 0.02 |
| $\beta_{N2}$ | The second order temperature parameter for N pool (1/C$^2$) | U(-0.05,0) | -0.24, -0.04, -0.03 |
| $\beta_{H1}$ | The first order temperature parameter for H pool (1/C) | U(0,0.2) | 0.002, 0.11, 0.20 |
| $\beta_{H2}$ | The second order temperature parameter for H pool (1/C$^2$) | U(-0.05,0) | -0.0001, -0.0014, -0.39 |
| Γ | The precipitation impact parameter for AWE pools (year/mm) | U(-2,0) | -0.93, -1.96, -1.34 |
| $\gamma_N$ | The precipitation impact parameter for N pool (year/mm) | U(-2,0) | -1.66, -0.32, -0.63 |
| $\gamma_H$ | The precipitation impact parameter for H pool (year/mm) | U(-10,-5) | -9.65, -6.15, -5.47 |
| $\phi_1$ | The first order impact parameter for size (1/cm) | U(-3,0) | -0.81, -1.41, -1.19 |





| $\phi_2$ | The second order impact parameter for size (1/cm²) | U(3,0) | 0.82, 0.25, 2.25 |
|---|---|---|---|
| r | The exponent parameter for size | U(0,1) | 0.83, 0.17, 0.49 |
| $w_{ED}$ | The leaching parameter for ED dataset | U(-1,0) | -0.08, -0.02, -0.05 |
| $w_{CIDET}$ | The leaching parameter for CIDET dataset | U(-1,0) | -0.03, -0.1, -0.08 |
| $w_{LIDET}$ | The leaching parameter for LIDET dataset | U(-1,0) | -0.08, -0.04, -0.02 |

Table 1: The parameters, prior distributions and initial values used in this calibration study. The initial values for the different chains were randomly drawn from the prior distribution (U: uniform). If the starting value is listed as a set value, then parameter was not varied in the calibration and the given value was used for all chains.






| Data | N | No. of species | Time range (a) | T range (°C) | P range (mm) | Elevation range (m) | Uncertainty used in calibration | Note | Reference |
|---|---|---|---|---|---|---|---|---|---|
| | | | | | | | | Mesh size (cm) | |
| Non-woody litter decomposition | | | | | | | | | |
| CIDET | 1259 | 10 | 0-6 | -9.8—9.3 | 261—1782 | 48–1530 | 100g | 0.25 x 0.5 | Trofymow 1995 |
| LIDET fine roots | 2608 | 4 | 0-10 | -7.4—26.3 | 150—3914 | 0–3650 | 200g | 0.055 x 0.055 | Gholz et al. 2000 |
| LIDET litter | 5900 | 29 | 0-10 | -7.4—26.3 | 150—3914 | 0–3650 | 200g | 0.055 x 0.056 | Gholz et al. 2001 |
| EURODECO | 2184 | 5 | 0-5.5 | 0.2—7 | 469—1067 | 46–350 | A:40g, W:10g, E: 20g, N:40g | 1 x 1 | Berg et al. 1991a, b |
| Hobbie | 192 | 4 | 0-5 | 6.7 | 3676 | 270 | 100g | 0.3 x 0.2 | Hobbie 2005 |
| Woody litter decomposition | | | | | | | | Diameter (cm) | |
| Finland | 1281 | 3 | 0-60 | 3.1 | 570 | na | 250g | 4.5—40.9 | Mäkinen et al. 2006 |
| SOC accumulation | | | | | | | | Soil depth (cm) | |
| Finland | 26 | | 5300 | 3 | 500 | 0 | NA | 0—30 | Liski et al. 2005 |
| SOC stock | | | | -26.9—28.0 | | | 7.5 kg | | |
| Global | 4113 | | | | 0–5663 | 0–3900 | | 0—100 | Zinke et al. 1986 |
| Finland | 30 | | | 3.2 | 681 | 115–180 | NA | 0—100 | Liski & Westman 1995 |
| Total | 17563 | | | | | | | | |

Table 2: The measurement data sets used in this research






| | DEzs MAP | DEzs G-R | DREAMzs MAP | DREAMzs G-R | AM MAP | AM G-R |
|---|---|---|---|---|---|---|
| αA | 0.51 | 1.01 | 0.46 | 1.06 | **0.53** | **1.15** |
| αW | 5.19 | 1.01 | 4.7 | 1.06 | **5.37** | **1.17** |
| αE | 0.13 | 1.01 | 0.11 | 1.04 | **0.13** | **1.23** |
| αN | 0.10 | 1.01 | 0.1 | 1.05 | **0.1** | **1.31** |
| αH | 0.001 | 1.02 | 0.002 | 1.02 | 0.002 | 1.07 |
| pWA | 0.5 | 1.00 | 0.5 | 1.03 | 0.50 | 1.04 |
| pWN | 0.16 | 1.01 | 0.17 | 1.08 | 0.16 | 1.06 |
| pEW | 0.99 | 1.02 | 0.97 | 1.09 | 0.98 | 1.09 |
| pH | 0.004 | 1.01 | 0.004 | 1.04 | 0.005 | 1.00 |
| wED | -0.19 | 1.01 | -0.18 | 1.03 | -0.19 | 1.03 |
| wCIDET | -0.03 | 1.01 | -0.02 | 1.04 | -0.02 | 1.04 |
| wLIDET | 0. | 1.01 | 0. | 1.01 | 0. | 1.04 |
| β1 | 0.16 | 1.00 | 0.17 | 1.08 | **0.16** | **1.30** |
| β2 | -0.002 | 1.00 | -0.002 | 1.07 | **-0.002** | **1.48** |
| β1N | 0.17 | 1.00 | **0.18** | **1.12** | 0.19 | 1.28 |
| β2N | -0.005 | 1.00 | **-0.005** | **1.13** | -0.006 | 1.38 |
| β1H | 0.07 | 1.02 | **0.07** | **1.15** | 0.07 | 1.55 |
| β2H | 0. | 1.02 | **0.** | **1.18** | 0. | 1.28 |
| γ | -1.44 | 1.01 | -1.66 | 1.09 | **-1.58** | **1.30** |
| γN | -2.0 | 1.04 | -2.0 | 1.07 | **-1.99** | **1.19** |
| γH | -6.9 | 1.01 | -5.78 | 1.09 | **-8.56** | **1.30** |
| φ1 | -2.55 | 1.01 | -2.32 | 1.03 | **-2.64** | **2.88** |
| φ2 | 1.24 | 1.01 | 1.18 | 1.05 | **1.32** | **2.69** |
| R | 0.25 | 1.01 | 0.25 | 1.05 | **0.25** | **2.08** |





Table 3 The estimated parameter value MAPs and Gelman-Rubin coefficients for DEzs, DREAMzs and
AM for global calibration against all data streams. Parameter that did not pass the G-R test are
bolded.



| Parameter | Global | CIDET | LIDET | ED |
|---|---|---|---|---|
| αA | 0.51 | 1.33 | 0.60 | 0.58 |
| αW | 5.19 | 9.95 | 9.40 | 5.74 |
| αE | 0.13 | 0.14 | 0.00 | 0.17 |
| αN | 0.10 | 0.02 | 0.08 | 0.10 |
| αH | 0.0015 | - | - | - |
| pWA | 0.50 | 0.59 | 0.59 | 0.41 |
| pWN | 0.16 | 0.08 | 0.06 | 0.16 |
| pEW | 0.99 | 0.26 | 0.42 | 0.97 |
| pH | 0.0042 | - | - | - |
| wED | -0.19 | - | - | 0. |
| wCIDET | -0.03 | 0. | - | - |
| wLIDET | 0. | - | 0. | - |
| β1 | 0.16 | 0.19 | 0.13 | 0.19 |
| β2 | -0.002 | -0.003 | -0.001 | 0. |
| β1N | 0.17 | 0.19 | 0.19 | 0.20 |
| β2N | -0.005 | 0. | -0.003 | 0. |
| β1H | 0.07 | - | - | - |
| β2H | 0. | - | - | - |
| γ | -1.44 | -0.46 | -1.74 | -0.88 |
| γN | -2.0 | -1.2 | -0.06 | -1.57 |
| γH | -6.9 | - | - | - |
| ϕ1 | -2.55 | - | - | - |
| ϕ2 | 1.24 | - | - | - |
| r | 0.25 | - | - | - |

Table 4 Estimated parameter MAP values with DEzs approach using all the datasets opposed to the individual litter decomposition experiments in the calibration





| Validation dataset | CIDET calibration | LIDET calibration | ED calibration | Global calibration |
|---|---|---|---|---|
| CIDET | **109.0** | 128.8 | 226.4 | 115.5 |
| LIDET | 224.3 | **168.8** | 345.4 | 199.9 |
| ED | 49.5 | 55.0 | **35.5** | 40.3 |
| Hob3 | 133.8 | 126.6 | 367.0 | **110.0** |

Table 5. The RMSE values for the different validation datasets when the model is ran with the MAP values from the calibrations done with the different datasets. As with the measurements, the RMSE unit here is grams. Lowest RMSE for a particular dataset is bolded.





| Site ID (Dominant tree species; Number of plots) | Averaged SOC (Standard deviation) | Projected steady state SOC |
|---|---|---|
| CT_SP (Pine; 5) | 5.78 (0.97) | 5.82 |
| VT_SP (Pine; 7) | 5.73 (0.71) | 7.39 |
| VT_NS (Spruce; 2) | 6.86 (0.67) | 8.78 |
| MT_SP (Pine; 4) | 6.89 (1.93) | 8.80 |
| MT_NS (Spruce; 7) | 8.61 (0.84) | 9.26 |
| OMT_NS (Spruce; 5) | 9.6 (2.2) | 10.26 |

Table 6: Both averaged measured SOC and projected SOC values for forest plots in Hyytiälä, Finland
classified by measurement site. All the units are in kgC m$^{-2}$.





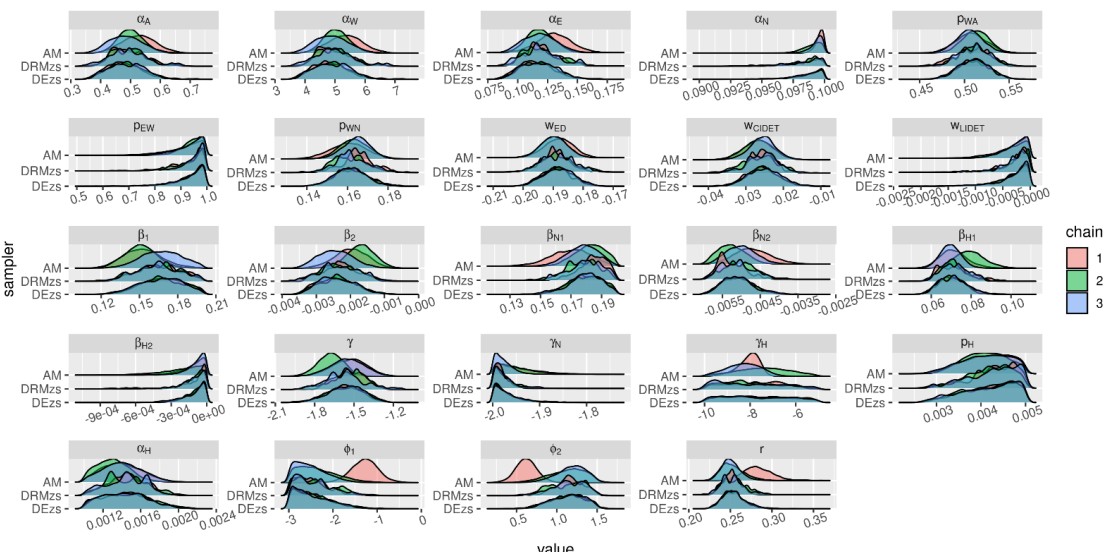

Figure 1: The global calibration results with the different calibration methods.




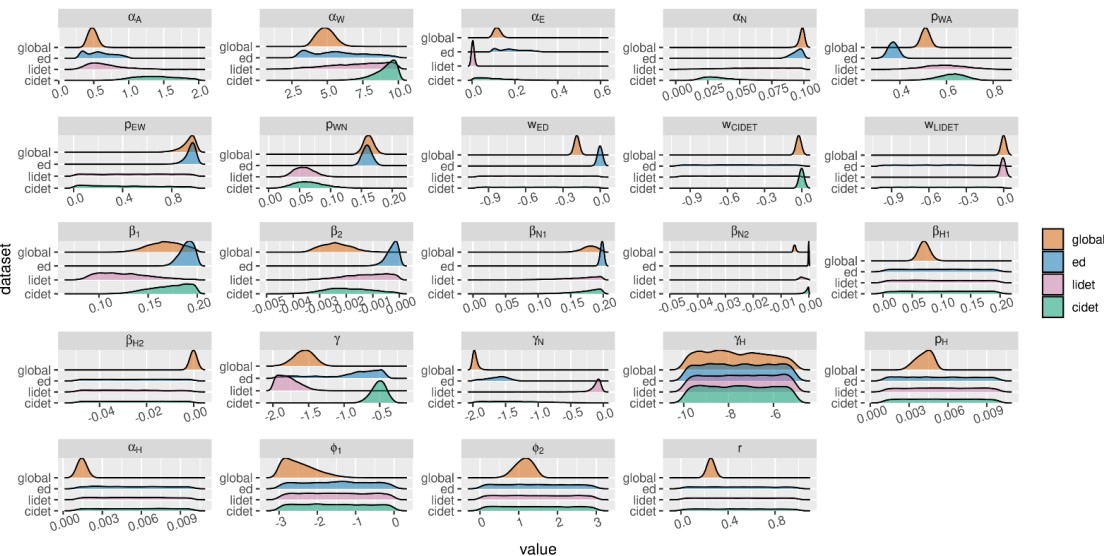

Figure 2: The estimated parameter distributions using DEzs with different calibration data sets.



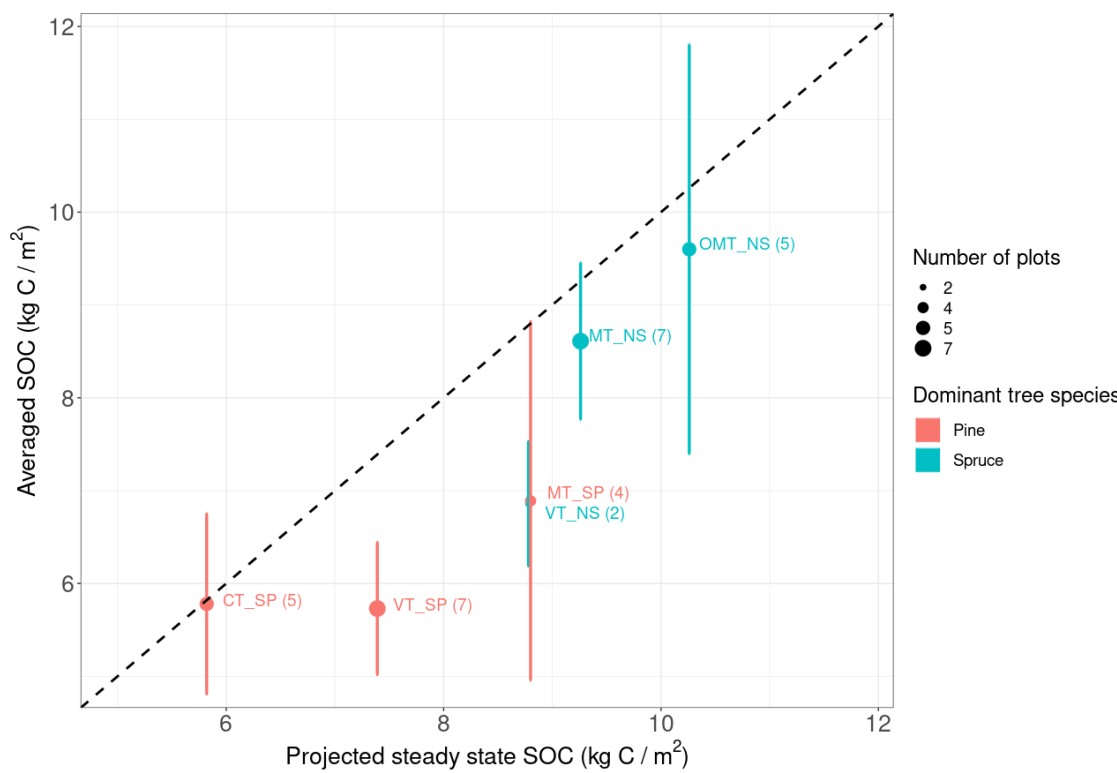


Figure 3: The projected steady state SOC compared to the averaged measured SOC values in plots from multiple measurement sites.



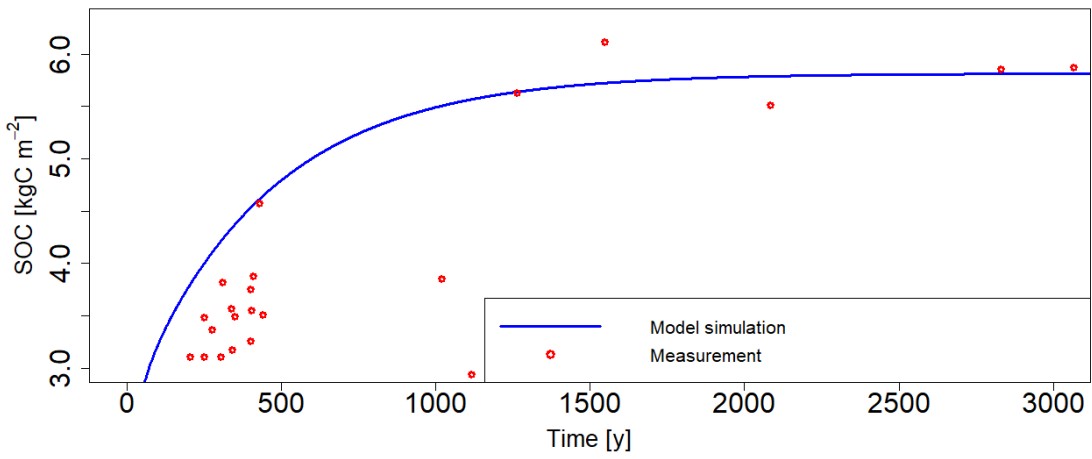


Figure 4: Measurement (Red dots) and model (Blue line) based projections of SOC accumulation on the Finnish coast after the end of ice age.



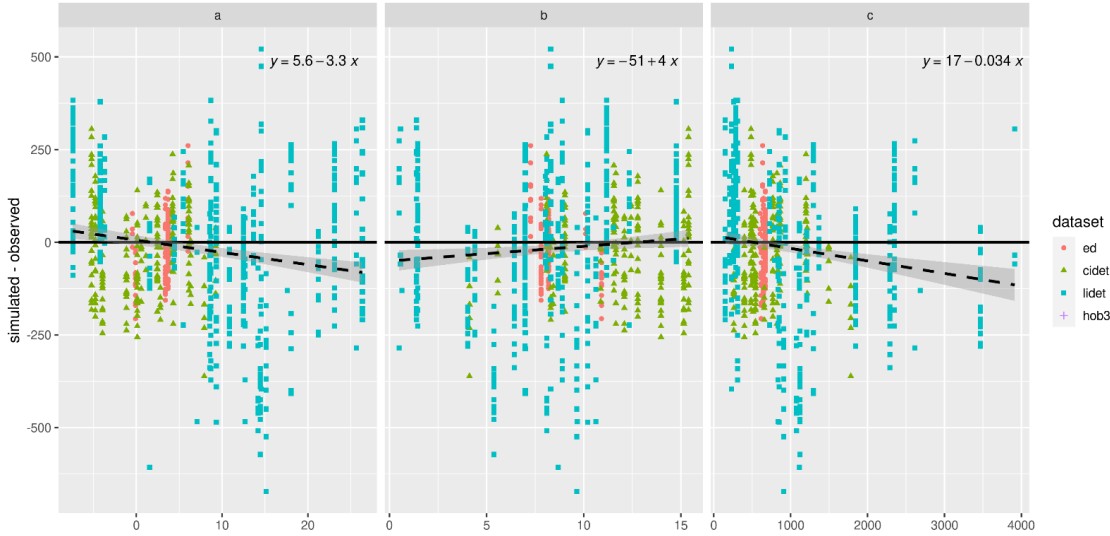

Figure 5: Residual analysis between simulated and observed carbon remnant on a) mean
temperature (C), b) temperature variation (C) and c) total precipitation (mm y$^{-1}$) at the
validation site.



a)

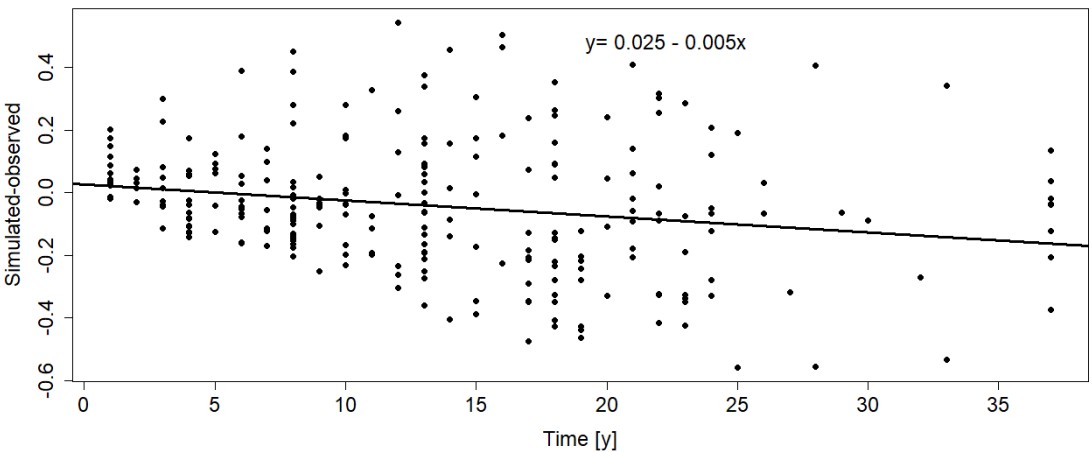


b)

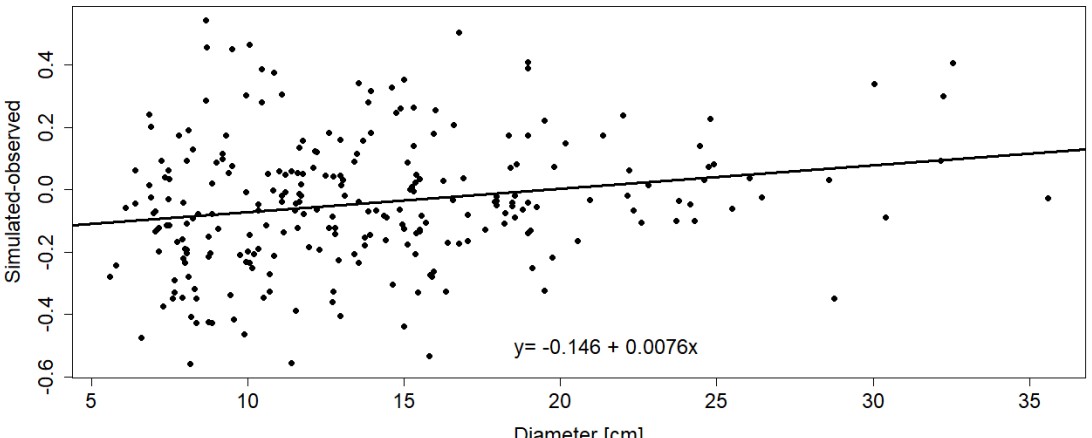

Figure 6: Residual analysis between simulated and observed carbon remnants of wood decomposition from Mäkinen et al. (2006) on a) decomposition time and b) diameter.






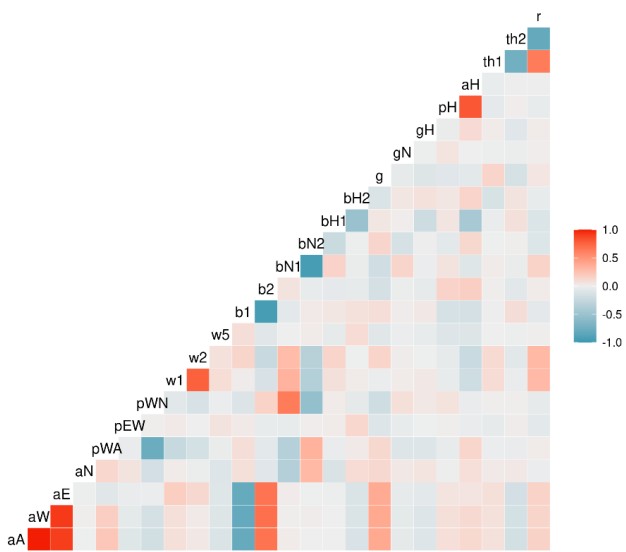

Figure 7: Parameter correlations for the global calibration with the DEzs algorithm.