# Peer review of "Calibrating the soil organic carbon model Yasso20 with multiple datasets"

_Geoscientific Model Development, 2021_

## Author Comment (AC1)

**Overall density of all chains**

---

## Author Response (AR1)

Dear Editor,

We are resubmitting the revised version of manuscript 'Calibrating the soil organic carbon model Yasso20 with multiple datasets.' Naturally we are deeply grateful to both reviewers for their time and effort in providing concrete suggestion on how to improve and clarify the manuscript. Hopefully these adjustments will be sufficient for the manuscript to be accepted for publication.

The remainder of this text provides point-to-point responses and details on how we have addressed the comments. The review comments are indicated in blue and our responses with black coloured text with indentation. The line numbers here refer to the lines in the change-tracked version of manuscript.

As a note, the Yasso07 results here are different from those in the original response to the reviewers. When doing a final confirmation of the code and the results, a flaw was found in the Yasso07 implementation that changed the results for it. The text has been changed to reflect the new results.

Dr. Toni Viskari

Senior Researcher

Finnish Meteorological Institute

toni.viskari@fmi.fi

Reviewer 1:

We thank Reviewer 1 for their feedback and have tried to strengthen the arguments presented in the manuscript to respond to the provided criticism.

"1. Line 85-86, the authors mentioned that they present a new model formulation and a calibration protocol. Please elaborate what is the "new" here, and please explain what the authors meant "calibration protocol"."

We expanded to be more descriptive of the changes. The manuscript now reads from line 96:

'In this study, we built upon previous Yasso developments to present a model formulation that expanded on how the environmental drivers affect the decomposition. The data used to calibrate the model is the same for both versions with the exception of the measurement data regarding long-term carbon allocation. For Yasso07, a time series dataset from Southern Finland while for Yasso20, approximated steady state SOC measurements from across the world was used to constrain the relevant parameters. Additionally, we use a more advanced model calibration method in association with a stricter protocol on what kind of data points were used for calibration and an open source R package for data inclusion, repetition and reproduction of calibration. The model and produced parameter set will refer to as Yasso20 hereafter.'

"2. Line 88 two "not only". Line 92, maximas-->maxima."

Corrected both points.

"3. Line 91-95, I do not think these two are hypothesis, they are more like facts to me. It is well-known that advanced parameter optimization methods (e.g., global parameter optimization methods, Markov chain Monte Carlo methods that can sample multi-modal distributions) perform better than the simpler methods in finding likelihood maxima. Additionally, it is also known that the models calibrated against multiple datasets perform better than those calibrated with a single dataset as long as the uncertain parameters are sensitive to the calibration data and the calibration data provide information to constrain the uncertain parameters."

In the first hypothesis, our intended point was not that the more advanced methods are just performing better, but they are particularly required in this system. Otherwise the given starting point will strongly affect the resulting estimated parameter sets. We have expanded on that part in the revised manuscript as reproduced below. The second hypothesis, however, we did not consider as obvious in this case as we are using data from very different datasets that have their own internal elements that cause issues in combining them. For example, comparing information from litter bag experiments isn't straightforward as the testing setup and bag properties will affect the decomposition in ways that cannot be modelled at this point. Hence here we have to include a crude leaching term to allow combining the information from the different experiments. Yet despite the assumptions required by this leaching term, the resulting parameter set will not be as deeply affected by the other error sources associated with a single dataset. Thus our hypothesis is that even despite these requirements, the estimated parameter set will still perform better as we see when examining how the various parameter sets perform with the independent Hobbes3 dataset. The new hypotheses, with the order of the hypotheses changed and a new one (regarding model comparison to previous version, which was implicit in our study) added to address suggestions from the second reviewer, are written from line 109 as follows:

'Due to the nature of the available SOC related datasets we hypothesize: I) the SOC model performs better globally if multiple datasets are simultaneously used to constrain it compared to a SOC model calibrated with an individual dataset despite the numerous assumptions required for combining the different information, II) the likelihood space created by these multiple datasets is uneven with multiple maxima to the degree that more advanced parameter methods are necessary for the end result not to be dependent on the starting point, and that

III) These changes in the model formulation and the calibration protocol and the expanded model formulation they allow for will improve how the Yasso model projections performance compared to the previous model version.

"4. Line 433-435, I assume the authors gave all the calibration data equal weights; the imbalanced data size may dilute the influence of Hobbie3 on the parameter estimation. How about giving different datasets different weights in model calibration?"

The Hobbie dataset is a litterbag experiment similar to those included in the CIDET and LIDET databases. While it does provide more specific information on the decomposition at that specific location, there's no reason to think it would give any more insight into how litter decomposes on a global scale than the data in the CIDET and LIDET datasets. As such, there is no real justification for giving it more weight. In other words, this is not an issue of a more abundant data type overwhelming a rarer data type, but simply having less of the same data type. We tried to clarify this point and add the re-weighing as a potential another tool to approach the general issue, with the paragraph starting from line 514 now reading:

'The inclusion of the Hobbie3 dataset did not meaningfully impact the calibration results (Not shown), which is reasonable considering how small that litterbag dataset (N=192) is compared to the totality of the other datasets (N=~17 000 of which $N_{litterbag}$ = ~12 000) being used in the calibration. This indicates that due to the sheer size of the global calibration data set, smaller local data sets cannot effectively be used just by adding it to the joint calibration process. Additionally, while the smaller datasets such as the Hobbie3 datasets contain site specific information, they are similar measurements as the ones within CIDET and LIDET and, thus, there is no reason to believe they would provide additional insight to the global application. There are other options, though, by either using the globally estimated parameter ranges as the priors for a calibration with the local data, re-weighing the different datasets based on expert opinion (Oberpriller et al., 2021) or employing a hierarchical calibration approach (Tian et al., 2020, Fer et al., 2021), but the impact of these approaches should be separately researched and tested. Our study still successfully provided a global parameter set that increases the applicability of Yasso model and informs global SOC estimates.'

"5. Subsection of impact of prior parameter information: in Bayesian calibration, the prior parameter distribution matters, not only the distribution type but also the prior parameter range. I think a global sensitivity analysis is necessary before the model calibration. At different portion of the parameter space, the parameters may have different sensitivity to the calibration data."

We acknowledge the importance of the prior range which is why we have used very loose (wide) priors to the parameter values and were careful to be certain that the initial values were sampled across the given space. The calibration results were also analyzed from this perspective as they would show how the estimated parameter values are affected by different regions in the parameter space. As we are not fixing any of the parameters, except the p values that end up too close to 0 or 1 in value, the calibration algorithms have the capability to fully explore the sensitivities of the different parameters.

Reviewer 2:

"Viskari et al. present a (re-)calibration of a slightly modified Yasso SOC model. The authors highlight in particular that calibrating to multiple data streams lead to better global model performance, and that the calibration was more successful with advanced population-based MCMC algorithms, compared to more traditional Metropolis-Hasting samplers.

While I agree with the remarks of my colleague (Rev 1) that the results regarding the superiority of multiple constraints and population MCMCs by themselves are rather unsurprising, I come to a different assessment regarding the overall novelty of this study. I see the issue highlighted by Rev 1

mainly in the presentation of the results, which concentrates in my option too strongly on generic technical aspects of the calibration, instead of stressing the improvements to the Yasso model that are facilitated by this calibration.

More specifically, my understanding is that the model used in this study is a previously unpublished improvement of the popular Yasso model that is calibrated and to some extent also validated in this study. To me, this seems valuable / novel, but this value would be more easily seen if the authors could better highlight the resulting benefits for SOC modelling. Moreover, if model improvements take a more central role of this paper, I would re-consider the decision to not compare the performance the new (calibrated) Yasso20 model with the older Yasso07 model – it seems to me that an improvement in the performance of the model would be a great argument to counter the novelty concern of Rev 1. "

Thank you for the constructive feedback and we have done our best to improve the manuscript based on it. The point about the Yasso07 comparison is valid and a straight-forward inclusion with the currently suggested changes in the manuscript. One of the initial challenges with the comparison was that the Yasso07 calibration data overlapped with the validation data used here as it used all the ED/CIDET/LIDET litterbag data and long-term SOC component of Yasso07 is calibrated with the time series data in Figure 5. The latter uses driver data from Southern Finland, which means that it is specifically calibrated for Hyytiälä conditions. However, Yasso20 actually produced SOC projections closer to the measured values with the Hyytiälä data, strengthening the argument concerning the improved calibration protocols having a positive impact on the model performance.

In order to address this, we have made the following changes to the manuscript:

In the introduction, we expanded the explanation of the differences between Yasso07 and Yasso20. From line 96, the manuscript now reads:

'In this study, we built upon previous Yasso developments to present a model formulation that expanded on how the environmental drivers affect the decomposition. The data used to calibrate the model is the same for both versions with the exception of the measurement data regarding long-term carbon allocation. For Yasso07, a time series dataset from Southern Finland while for Yasso20, approximated steady state SOC measurements from across the world was used to constrain the relevant parameters. Additionally, we use a more advanced model calibration method in association with a stricter protocol on what kind of data points were used for calibration and an open source R package for data inclusion, repetition and reproduction of calibration. The model and produced parameter set will refer to as Yasso20 hereinafter. Our redesigned calibration protocol leverages BayesianTools R-package (Hartig et al., 2019), an open source general-purpose tool for Bayesian model calibration. Using BayesianTools in our workflow, we not only established a more reproducible and standardized application of Yasso20 calibration, but also leveraged interfacing with multiple calibration algorithms and examined the role of the calibration method.'

The hypotheses were rewritten to include the Yasso07/Yasso20 comparison with the new hypotheses starting from line 109 being:

'Due to the nature of the available SOC related datasets we hypothesize: I) the SOC model performs better globally if multiple datasets are simultaneously used to constrain it compared to a SOC model calibrated with an individual dataset despite the numerous assumptions required for combining the different information, II) the likelihood space created by these multiple datasets is uneven with multiple maxima to the degree that more advanced parameter methods are necessary for the end result not to be dependent on the starting point, and that III) These changes in the model formulation and the calibration protocol will improve how the Yasso model projections performance compared to the previous model version.'

We added a short paragraph in section 2.2 to clarify what were the differences in data used to calibrate Yasso07 and Yasso20.  From line 250, the section now reads:

'The same litterbag and woody data were used to calibrate both Yasso07 and Yasso20. However, Yasso07 H pool parameters were not parameterized with the Oak Ridge data. The sole exception regarding the litterbag data is that the whole ED dataset was used in Yasso07 calibration while in Yasso20 we removed decomposition data from manipulation experiments. Instead, the chronosequence data from Liski et al. (1998) was used in its calibration with climate and litterfall drivers derived from Southern Finland conditions (Tuomi et al., 2009). As already established, this dataset was not used in Yasso20 calibration and was only applied as a validation dataset.'

We also created a new section to explain how the Yasso07/Yasso20 comparison shall be done. This section starts from line 375 and reads:

**'2.5 Yasso07/Yass020 comparison protocol**

During the calibration of Yasso07, there was no separate validation data set aside for the CIDET, LIDET and ED and all the data was used for the parameterization. Because of that we do not use those validation datasets for the model performance comparison. Instead, only the Hobbie3 validation dataset and the Hyytiala plots are used to determine if there is any notable improvement in Yasso20 performance with them compared to Yasso07. For the litterbag data, the comparison shall be the RMSE while for the Hyytiala plots how the model projected steady state SOCs compare to the measured plot values.

To assess the differences in the model over long-term decomposition, both models were used to model the decomposition of a hypothetical straw litter (A=620 g, W=50 g, E=20 g, N=310 g) over a 100-year time period. This is not based on any measurement time series and is purely a synthetic test.'

The results from the Yasso07/Yasso20 comparison revealed that while superficially the projections for both short-term decomposition time series and steady state SOC were similar, there were fundamental differences in the model dynamics and the states they produce. In Yasso07, the N pool decomposition is much slower than in Yasso20, which results in the majority of the decomposed carbon accumulating in the N pool and staying there for a long period of time. As a consequence, there is similarly not much allocation in the long-lived H pool with the soil SOC and the carbon compounds there are much more short-lived. We explain this in the results section starting from line 456 as follows:

'When examining how Yasso20 performs relative to Yasso07, the RMSE for Yasso07 projections is 118.2 grams compared to the Yasso20 RMSE of 110 grams. With the Hyytiälä forest plot measurements (Table 4), in all plots Yasso07 overestimated the SOC by at least 3 kg C m⁻² more than Yasso20. However, when examining the distribution of carbon into different pools in these steady states (Not shown), more meaningful differences were revealed. For Yasso07, only ~37 % of the SOC was in the long-lived H pool while ~50 % of the carbon was in the N pool. By comparison, with Yasso20 projections ~54 % of the carbon is in the long-lived H pool and ~27 % in the N pool.

The hypothetical straw litter decomposition (Figure 6) shows that while the total carbon remainder for the two models are close to each other for the first 10 years, after that there is a clear divergence between the model projections with Yasso07 having higher remaining carbon than Yasso20. More detailed inspection of the results (Not shown) found that this difference was due to the N pool decomposing at a much slower rate than in Yasso07 than in Yasso20. This also causes less carbon to accumulate in the H pool in Yasso07 than in Yasso20 with the latter having approximately twice as much carbon in the H pool than the former after 50 years. When repeated with warmer climate drivers (Not shown), Yasso07 time series projection decreases at a faster rate than Yasso20 time series projection.'

In addition we adjusted Table 4 to also include the Yasso07 steady state SOC projections:

| Site ID (Dominant tree species; Number of plots) | Averaged measured SOC (Standard deviation) | Yasso07 projected steady state SOC | Yasso20 projected steady state SOC |
|---|---|---|---|
| CT_SP (Pine; 5) | 5.78 (0.97) | 8.32 | 5.82 |
| VT_SP (Pine; 7) | 5.73 (0.71) | 10.61 | 7.39 |

| | | | |
|---|---|---|---|
| VT_NS (Spruce; 2) | 6.86 (0.67) | 13.06 | 8.78 |
| MT_SP (Pine; 4) | 6.89 (1.93) | 12.78 | 8.80 |
| MT_NS (Spruce; 7) | 8.61 (0.84) | 13.87 | 9.26 |
| OMT_NS (Spruce; 5) | 9.6 (2.2) | 15.47 | 10.26 |

The new added Figure 6 is below:

a)

[Figure]

b)

[Figure]

Figure 6: The remaining decomposing carbon for a hypothetical straw litter in Hyytiälä, Finland climate condition when simulated with Yasso07 (solid red) and Yasso20 (dashed blue) with a) 20 year and b) 100 year time window.

Finally we focus on the comparative performances and the issues raised during the results analysis in a specific section of the Discussion section. Starting from line 587, the manuscript now reads:

**'How Yasso20 performs in comparison to Yasso07**

When comparing the litter bag validation dataset performances of Yasso07 and Yasso20, there is an improvement with Yasso20 even though both models have been calibrated largely with the same litterbag data. This underlines that the added model detail and reconsidered calibration process have a positive impact on the model projections. What is more striking, though, is that Yasso20 does perform better across the board with the Hyytiälä SOC data than Yasso07 where the latter model's long term SOC component was calibrated with Finnish conditions. This result argues that while local calibration data is important, even for those specific locations there could be a benefit in including global data in the calibration. These results validate the third hypothesis concerning the impact of the presented improvements on model performance.

A more thorough analysis of the model projections revealed a more fundamental difference in the model dynamics than initially indicated by the comparison datasets. In Yasso07 the N pool decomposes much slower, which impacts the rest of the decomposition dynamics and causes less long-lived H pool carbon to be formed during the soil decomposition. As a consequence of differences in the calibration procedures and the resulting model versions, Yasso07 projects higher SOC values than Yasso20 with the same input values and these model versions would also react differently to changes in climate conditions and litter input.

The Yasso07 dynamics are most likely due to a combination of multiple reasons which highlights the complicated process of SOC model calibration. As Yasso07 was calibrated in segments, the woody decomposition parameters were calibrated after the AWENH pool parameters were determined from the global litter bag experiments and Finnish SOC measurements. When looking at the calibration results from individual datasets (Figure 2) there are parameter sets there which have similarly low decomposition rates for N pool as Yasso07. Depending on how the different measurement datasets were weighed, it might be that those datasets that favored slower N pool decomposition had more impact than with Yasso20 calibration. Finally, in Yasso20 the climate driver parameters are different between the AWE and N pools and while the temperature terms are close to each other, the precipitation terms do differ from each other while in Yasso07 they would be the same. This would affect the Yasso07 dynamics during calibration. The calibration is made even more vulnerable to all these factors because a  vast majority of the litter bag data used here is from the first six years of decomposition where Yasso07 and Yasso20 are very close to each with regard to total carbon remaining (Figure 6). In such a situation it is very possible that less developed  calibration protocols can lead to unrealistic system dynamics that still appear to produce good results within limited time windows.'

"Regarding the comments of Rev 1 that a global SA is necessary before calibration, and that the likelihood should be weighted: I think both are good points that should be considered, but I also think that the approach taken by the authors is not necessarily wrong. Performing an SA prior to calibration has the main purpose of reducing the number of parameters in the MCMC, which speeds up calculations. If the authors manage to calibrate their model despite not performing an SA, I don't see a problem. The topic of weighting is a bit more tricky: statistically, arbitrarily-reweighting data is difficult to defend (although this is widely applied). As we show in Oberpriller et al. (2021) Ecology Letters, if the model is 100% correct, weighting has no benefits to the calibration. However, as we show in the same paper, if there are systematic model errors, weighting can be beneficial for obtaining reasonable fits when data are strongly unbalanced and weighting is done appropriately. As these conditions may be met here, I would agree that the authors could experiment with whether re-weighting the data improves model performance, however, I wouldn't say that re-weighting is a categorically better or absolutely needed. "

We completely agree with the points raised here. From the perspective of our work, the challenge of the re-weighing is that there is no real support that one of the datasets used should be weighed more. For example, all the litterbag datasets are collections of litterbag experiments with some such

as LIDET having more data points than others such as CIDET. However, there's no reason to think that for global performance the CIDET data is more representative which would support giving it more weight. It could be argued that CIDET experimental setup is more reliable, but that, however, is already taken into account with the uncertainties. We have tried to expand on our explanation in relation to this with the relevant line-on-line comments below.

"In summary, I think that this is solid model calibration study. There are a few minor technical issues that can be found below, the most critical is probably that I would recommend also calibrating the error terms in the likelihood. Also, the issue of re-weighting could be considered. To clarify the novelty of the study, I would recommend to re-structure the presentation around the overall goal of model improvement and useability (e.g. that you show how to perform quick / efficient calibrations for this particular model). The question of multiple data-streams and sampler choice is interesting, but with the focus on model improvement, it would take a more supportive role in the overall story."

In response to the general comment about the model improvement and usability, in our overall revisions we have focused on highlighting the model comparison as well as strengthening the hypothesis presented in the manuscript.

Regarding calibrating the error terms, we touch on this in the more specific comment made below, but justo quickly mention here, we now tested calibrating the error terms in the likelihood, and it does not result in overall improved model performance with the validation datasets, although it does not function worse either.

"Also, while being perfectly intelligible, I believe that the general conciseness / flow of the text could still be improved."

We believe the suggested revisions have improved the flow of the text and made it more concise. Before resubmitting we will, though, go through the manuscript a final time to try to find parts which could be further streamlined.

In the more specific correction suggestion responses below, the line numbers given correspond to where the text was altered/added in the initially submitted manuscript.

"Title: consider erasing „the impact of" (conciseness)?"

Agreed, removed the 'impact of' part of the title and the new title is 'Calibrating the soil organic carbon model Yasso20 with multiple datasets'.

"10: "tools in determining" -> to determine?"

Changed it to 'tools for assessing'.

"16 erase "the""

Erased.

"21 In terms of the logical flow, I would recommend starting with the topic of soils here, as this would allow you to move more naturally to the models at the end of the paragraph (as opposed to the current structure, which is going models -> soils -> models)"

Thanks for this suggestion, switching the topics around did improve the flow. The first paragraph of introduction starting from line 21 now reads:

"Soils are the second largest global carbon pool, hence even small changes in this pool impact the global carbon cycle (Peng et al. 2008). However, Soil Organic Carbon (SOC) and associated changes are difficult and laborious to measure (Mäkipää et al., 2008).  They can also vary drastically over space due to differences in litter fall, site and soil type as well as climate (Jandl et al., 2014, Mayer et al., 2020). Hence, SOC models are important tools for estimating current global soil carbon stocks and their future development (Manzoni and Porporato, 2009).  Numerous SOC models have been developed in the past decades (Parton et al., 1996; Cammino-Serrano et al., 2018; Thum et al., 2019) to quantify the global SOC stocks and estimate the effects of different drivers, such as changing environmental conditions, on SOC stocks (Sulman et al., 2018, Wiesmeier et al, 2019)."

"21 FOR estimating?"

Corrected.

"42 The third challenge feels a bit like an add-on. More generally, I wasn't convinced about the sense of classifying these three distinct challenges, because they are (as you note) connected. Maybe it would be easier to state something along the lines that there is evidence that we should add more complexity to the models (e.g. nonlinearities), but that empirical (data availability, spatial variation) and methodological challenges (data assimilation) have so far hindered successful expansions of model complexity."

An excellent suggestion and the paragraph starting from line 36 has been rewritten as:

'While majority of SOC models rely on linear equations representing the movement of C within the soil, there has been studies showing the need to represent at least some of the SOC processes such as the microbial influence by non-linear equations (Zaehle et al., 2014; Liang et al. 2017) or that the state structure of the model affects which kind of data can be used to calibrate it (Tang and Riley, 2020). More complicated SOC models addressing these arguments have been developed, for example Millennial (Abramoff et al, 2018), and modules including additional drivers affecting the C pools have been included in existing SOC models, such as nitrogen (Zaehle and Friend, 2010) and phosphorus (Davies et al, 2016; Goll et al., 2017) cycles. Their implementation is hindered, though, by that detailed data is needed to constrain the model parameterization, but individual measurements campaign datasets are often limited in size and lacking in nuance of the SOC state (Wutzlerand and Reichstein, 2007; Palosuo et al., 2012). Consequently, multiple datasets representing different processes should be used to parameterize the models in order to capture the multitude of SOC dynamics, but combining observation datasets with varying spatial scales, measurement temporal densities, inherent assumptions and structural errors can cause issues with adequately incorporating all the information (Oberpriller et al., 2021). The chosen calibration methodology is additionally affected by the same issues based on its approach of fitting the data. '

"46 The logical flow / connection of this new to the last paragraph is hard to grasp. That you need multiple data streams was already stated in the last paragraph (challenge 2). Possibly, you could remove this from the previous paragraph and say here that calibration challenges could be addressed by combining multiple data streams. In this context, the comments in the intro of Oberpriller et al., 2021, https://onlinelibrary.wiley.com/doi/full/10.1111/ele.13728 may be of interest."

We removed the previous connecting sentence and now shift directly to discussing the different kinds of measurements used here. We also included a reference to the Oberpriller et al. (2021) into the previous paragraph.

"86 It sounds here as if you refer to the model + calibration protocol as Yasso20, but below (114) you refer to the model alone as Yasso20. I think you mean the latter, right?"

The original intent was that the parameter set produced by the calibration is also considered a part of Yasso20 and we clarified the text to better convey this. The manuscript now reads from line 96:

'In this study, we built upon previous Yasso developments to present a model formulation that expanded on how the environmental drivers affect the decomposition. The data used to calibrate the model is the same for both versions with the exception of the measurement data regarding long-term carbon allocation. For Yasso07, a time series dataset from Southern Finland while for Yasso20, approximated steady state SOC measurements from across the world was used to constrain the relevant parameters. Additionally, we use a more advanced model calibration method in association with a stricter protocol on what kind of data points were used for calibration and an open source R package for data inclusion, repetition and reproduction of calibration. The model and produced parameter set will refer to as Yasso20 hereinafter.'

"115 I realize this information is provided later, but I think it would help the reader at this point to have one sentence that clarifies how Yasso20 differs from Yasso07. Also, clarify in the description that follows whether descriptions refer to the Yasso07 or Yasso20."

Thanks for the suggestion. We now added the sentence and the requested clarification. The manuscript now reads from line 143:

'Yasso20 is the next version of Yasso (Liski et al. 2005) and Yasso07 models (Tuomi et al., 2009, 2011b) and continues to build on these same assumptions. The main formulation contribution in Yasso20 compared to the previous versions is the added nuance in how climate drivers affect the different pools, which in turn is possible here due to the improved calibration scheme. For the purposes of the calibration here, another assumption was necessary: 5) The most stable soil carbon compounds are only formed in the soil as a result of bonding with mineral surfaces (Stevenson, 1982). The following model formulations apply for Yasso20.'

"164 On the github repo that you link, there also seems a Yasso15?"

Yasso15 is the first attempt at a more expansive Yasso calibration which simultaneously used all the datasets in this manuscript. We weren't quite satisfied by all the aspects of the calibration there, which led to Yasso20 here, which does perform better with the different validation than Yasso15 did. However, while Yasso15 has been used in some publications such as Viskari et al. (2020), it was never published in detail which is why Yasso07 is more used and why we are focusing on Yasso20 being the next step from Yasso07.

"Table 1,3,4: These are quite long, maybe some of these could be combined, presented visually or moved to the supplementary? I think the main text should concentrate on the central messages of the paper."

Fair point on the tables. Table 1 presents what the different symbols in the figures represent and is a list of the parameters being calibrated, so we hesitate to remove that from the manuscript. The information in Table 3 and 4, though can be seen to a degree in Figures 2 and 3, so we will move those two Tables to supplemental material. Additionally while we kept Table 1 in the manuscript, it is not a strong position and we will move it to Supplemental material if requested again.

"221 It seems that you assume in this section (I also looked at the code to make sure) that data uncertainties (i.e. sd in your likelihood) have to be fixed a priori from the data. This, however, is rarely a good idea. Even if you know the observation error perfectly, there can be other reasons for your model to deviate from the observed data (e.g. model error, or some variability in the environment that has nothing to do with the observation process). Consequently, if you fix the sd in the likelihood based on your observation uncertainty, you will get wrong (typically too narrow) posterior distributions. I would highly recommend calibrating with variable sds. If you want, you can

set priors to reflect your data uncertainties, but you should give the calibration a chance to correct those if necessary."

As recommended, we repeated the calibration with the uncertainties also being among the variables with the results below for a calibration run of 2.5M iterations. The only difference in how the uncertainties were used here in the likelihood calculation was that with the ED dataset where instead of using different uncertainties for all the AWEN pools, we only used one uncertainty for the whole dataset. This was because even with this approach and at this length, the calibration still had difficulties converging with the G-R values for aA, aW and r remaining above 1.1.

[Figure]

In these results are the estimated parameter uncertainties not only mostly similar to the ranges estimated with the fixed uncertainties, but with some like aA the uncertainty range is actually smaller. The system dynamic that the parameters settle down in these results raise some questions as the AWEN decomposition rates here are notably slower than with the fixed uncertainty calibration. Additionally due to the lack of convergence with the r parameter, it ends up with a parameter set where the simulations become unstable with small wood decomposition.

With the validation datasets, the performances were about even with a calibrated uncertainty parameter set performing better with some validation datasets than the fixed uncertainty and vice versa. The Hyytiälä data comparison could not be done due to the instability issues with woody decomposition. The RMSE comparison of the validation results are below.

| Validation dataset | Fixed uncertainties | Estimated uncertainties |
|---|---|---|
| CIDET | 115.5 | 118.8 |
| LIDET | 199.9 | 181.1 |
| ED | 40.3 | 28.8 |
| Hob3 | 110.0 | 119.7 |

While the RMSE does improve considerably with LIDET and ED, there are also some questions relating to that as they both are noisier than the CIDET dataset, but the calibration actually reduces their uncertainties compared to the fixed uncertainties. With ED this is without a doubt connected to the single uncertainty used which leads to more weight given to A and N pools. Additionally the reduced uncertainty calibrated for the global steady state is noticeably small considering that it is an unreliable dataset for multitude of reasons.

Overall the calibrated uncertainty appears to perform as well as the fixed uncertainty, although the issues related to woody mass decomposition would still need to be solved. While we fundamentally agree with the reviewer about the fact that even if we know the observation error perfectly, there can be other reasons for the model to deviate from the observed data (and we also agree that they are important to study), we originally went with our approach of fixing the uncertainties as the interpretation of the calibrated uncertainty then becomes obscured where the discrepancy is now lumped with data error. As fitting the uncertainties didn't affect overall performance and general conclusions much, and fixing the uncertainties have an easier explanation to be rough approximations, we are considering to keep our original approach in the main text and put this exercise in the supplement unless the reviewer strongly disagrees.

"290 In general, the calibration part provides very little info on the most crucial part of the calibration, which is the likelihood that you calibrate. The details on the algorithms are useful, but this could also go in the appendix."

This was an oversight on our part and thank you for pointing out the lack of information regarding the likelihoods used. We still kept the algorithm introduction in the text as while it is generic, it is details on the implementation. We did, though, add a paragraph discussing the likelihood approach used here.

To expand on the chosen approach on the likelihoods in response to the question, we used a very simple approach in that the applied uncertainties are normally distributed, the given uncertainty is static in time and they are independent of each other. Naturally based on our current knowledge, these assumptions do not hold for this data, but the challenge is that there isn't sufficient information on what additional uncertainty constraints should exactly be. Not only that, but the framework we would implement in calculating them from the current data would have a direct impact on what kind of likelihoods we would get.

As an example, we use a static uncertainty for each decomposition experiment dataset regardless of measurement time. The thought process for this is that while the absolute uncertainty would be larger at the beginning of the time series as there is more litter remaining, over the time series the relative uncertainty would actually increase due to a multitude of factors. There is, however, no published approximations on how the relative error changes over time and it even actually depends on the dataset as with CIDET the bags are on a platform while with ED they are on the surface which would impact on how the relative error develops in both experiments. Another example would be autocorrelation as in a decomposition time series it is expected that if the model projection underestimates the amount of carbon left after three years, it is more likely it will also underestimate the remaining carbon after four years. Due to the currently poorly understood dynamics, though, with litter decomposition bag data this assumption does not hold true for the start of the time series. As a consequence, the choice of time window over which the autocorrelation is calculated will affect the resulting autocorrelations.

In our view there were two equally justified approaches on how to handle this: Either apply a series of semi-reasonable assumptions on what those different uncertainty components are, or keep the system simple due to not really having that more accurate information. Regardless of the choice, it is

crucial to be clear on the limitations and the reasoning of it, which is why it is on us that we did not have noticed that we did touch on it. Of the two options, we chose the simplistic implementation not just because it was more straight-forward to implement, but also because, once more accurate information on the uncertainties becomes available, it is easier to assess the impact of it when adding it instead of changing its inclusion.

We added a paragraph at the end of section 2.3 to attempt to clarify the uncertainties starting from line 322:

'For the likelihood function we used a simple approach where the uncertainties are assumed to be normally distributed and independent of each other. In the litterbag experiments, because the absolute uncertainty remains the same over time while the amount of decomposing litter decreases, the relative uncertainty increases over time. There are error dynamics affecting the data in reality that are not accounted for here such as more nuanced time dependence of the uncertainties, uncertainty auto-correlation in a time series and non-normally distributed uncertainties. Due to not having reliable information to properly assess how these effects should be included into the likelihood calculations here, we chose the described basic approach. This is considered to make it more straight-forward to later add the missing uncertainty dynamics as approximations of them become available and examine how those inclusions affect the calibration results.'

"333 Different or not converged? It seems not converged, but then you shouldn't interpret it and just run it longer."

We agree with the reviewer about the misleading use of the technical term "convergence" here and thank the reviewer for pointing this out. We meant that the individual AM chains referred to in the manuscript here have converged (for the lack of a better word) and got stuck at the same values even if we run the calibration for even 3 million iterations, which would be twice the amount of iterations used in the manuscript. Where individual chains converge (or get stuck) is affected by the starting point, which in our argument shows how uneven the likelihood space for these datasets is and why using more advanced MCMC methods is paramount. We expanded the results to clarify this and now it reads from line 395:

'Closer examination of different chains, though, shows that while DEzs and DREAMzs converged to the same parameters, individual AM chains instead produced different parameter distributions and thus the calibration itself did not converge. The AM chain parameter distributions already settled into these distributions based on the initial parameter values given to them and even after doubling the number of iterations (Not shown) the distributions remained the same. In our view, this is indicative of what would happen if a simple single chain calibration was done with SOC models.'

'354 Via the arguments in Oberpriller et al., 2021 (cited above) these observations could be interpreted as a hint of systematic model / data error. Would you agree? Possibly to be added to the discussion.'

Agreed and we originally briefly touched on it when discussing the impact of the leaching term. We expanded that part of the discussion to better reflect this and it now reads starting from line 633:

'A further complication is that the differences in RMSE results (Table 3) suggest that there are systematic differences between the datasets resulting from various sources such as the experimental setup or environmental differences. As a consequence, calibrating with these kinds of datasets will result in systematic differences in model performance as established in Oberpriller et al. (2021) as can be seen in how CIDET/LIDET calibrated Yasso performs with the ED dataset and vice versa. By being a corrective term, the leaching factor introduced here will also reflect all those other elements causing the systematic differences, for example different mycorrhizal environments, instead of just being about the physical properties of the litter bag. Due to all these factors,the leaching impact needs to be further studied and the relevant equations need to first be formulated with experimental data specifically gathered for that purpose. There also needs to be additional work in trying to better quantify what those other systematic error elements are so that they can be better addressed.'

"375 Here, but also other sections: I think it would be helpful for the reader if you would re-state at the beginning of each result what the purpose / motivation for the respective result was. This section, for example, doesn't trivially connect to a research question of yours, nor is it mentioned in the methods that you would look at this, so you should give the reader a bit of context. I also wonder if the correlations wouldn't better be mixed with a discussion of the mean parameter estimates, which seems to be missing. Such a discussion would imo logically be better placed BEFORE the discussion of predictive performance, but after the discussion of the calibration performance (convergence)"

Excellent point and we have added/rearranged the Results section based on this feedback in the following manner. We still begin with the comparison of the different calibration methods as this justified our choice of the DEZs for the dataset experiment before moving on to presenting the parameter values together with the correlations, and finally we end with the validation/residual analysis. Furthermore in the validation part of the results we also compare the performance of Yasso20 to the performance of Yasso07 with the same validation datasets.

We added an introductory line starting at line 390 to establish what was examined in this first part

'The first step was to determine if there is a notable difference in how the different calibration methods perform with the global dataset.'

Then we rewrote the rearranged parameter value analysis as well as looking at the impact of the multiple datasets used for calibration. That section now starts from line 412 as follows:

'The next step was to examine how the use of multiple datasets simultaneously affected the calibrated parameter sets compared to when using only individual datasets for calibration. The parameter sets produced by the calibrations differ from each other to a meaningful degree in both the parameter mean value as well as the associated uncertainty range (Figure 2; supplemental table 5). Despite that, though, there are certain patterns in the parameter sets: The pool decomposition rate relationships remain the same in that W has the quickest turnover rate followed by A with N being the slowest to decompose. With the climate terms, both CIDET and LIDET calibrations have difficulties in settling on the climate terms while covering a multitude of different climate types while ED calibration, where the climate differences between measurement locations are minor, produces a relatively narrow climate parameter estimate. The global calibration, however, does clearly converge around certain climate parameters even if the uncertainty range remains wide. And even though the ED dataset has the most detail about the AWEN distribution, the AEW decomposition rates estimated based on it do not appear to converge with multiple peaks in the parameter distributions.

To further examine the parameter calibration results, we analyzed the correlations between different parameter values produced by the DEzs algorithm from the global calibration (Figure 3), which shows that the correlations are the strongest between processes affecting the same pools. The p-terms which had been set to 0 and 1 were excluded from the correlation analysis since they did not vary during the calibration. The AWE pools decomposition rates have strong positive correlations between the decomposition rates as well as with the climate driver terms affecting decomposition in them. Similarly, there are strong negative correlations between the temperature terms affecting the same pools and a strong positive correlation between the H pool terms. There are both strong positive and negative correlations with the size related parameters. While the exact correlation values changed depending on the calibration dataset, the general relationships remained similar (Not shown).'

We also added a sentence to line 438 explaining the final part of the results:

'The final step was to validate how the different parameter sets perform with separate validation datasets and determine if there are notable systematic errors with regard to the driver data.'

"385 As I said in my general comments, I believe you should put the model improvements (i.e. updated parameters, improved performance) in the center, and present the results about MCMC algorithms and multiple data streams rather as a byproduct."

We have adjusted the manuscript to bring the focus better on the model performance improvements. First we switched the order for and expanded the hypothesis starting from line 106:

'Due to the nature of the available SOC related datasets we hypothesize: I) the SOC model performs better globally if multiple datasets are simultaneously used to constrain it compared to a SOC model calibrated with an individual dataset despite the numerous assumptions required for combining the different information, II) the likelihood space created by these multiple datasets is uneven with multiple maxima to the degree that more advanced parameter methods are necessary for the end result not to be dependent on the starting point, and that III) These changes in the model formulation and the calibration protocol will improve how the Yasso model projections performance compared to the previous model.'

Then we begin the Discussion section with the model improvement and moved the smaller data set inclusion to a separate paragraph. Starting from line 495, the Discussion now begins:

'**The benefit of calibrating with multiple datasets**

Our results show that simultaneously using multiple datasets from different environments improves the general applicability of the SOC model even when having to use the simplistic leaching factor approach to be able to compare different litter bag datasets and lacking detailed uncertainty estimates, confirming our first hypothesis. This is in line with prior studies arguing for larger representation in the calibration data (Zhang et al., 2020). Furthermore, a more detailed analysis of different calibrations shows (Figure 2) indicates that the information from multiple datasets is in truth even necessary for the calibration as when calibrating only with one dataset, the decomposition parameter uncertainty ranges either were large or, in the case of the more nuanced EuroDeco dataset, don't even appear to converge.

When comparing the litter bag validation dataset performances of Yasso07 and Yasso20, there is a noticeable improvement with Yasso20 even though both models have been calibrated largely with the same litterbag data. This underlines that the added model nuance and reconsidered calibration process have a positive impact on the model projections. What is more striking, though, is that Yasso20 does perform better across the board with the Hyytiälä SOC data than Yasso07 where the latter model's long term SOC component was calibrated with Finnish conditions. This result argues that while local calibration data is important, even for those specific locations there could be a benefit in including global data in the calibration.'

"394 Maybe I missed it, but did you show that multiple maxima were the problem? In my experience, trade-offs between parameters are a far more common in the context that you consider here."

We thank the reviewer for pointing this out. This was a misuse of the terminology on our part where we presented the multiple maxima as a consequence from the parameter trade-offs. We now rephrased the sentence starting on the line 533 to read as follows:

'This supports our second hypothesis that more advanced calibration methods are necessary to better explore the likelihood surface and estimate SOC model parameters due to the trade-offs between the parameter values result in equifinality in the parameter space.'

"396 OK, but isn't that to be expected?"

It is. We now removed the sentence per the restructuring of the discussion where we touch on this now in the new first paragraph of the Discussion section. There we try to also highlight that it is not just that using multiple datasets improves the performance, but that it is necessary for parameter sets to actually converge.

"404 I also don't know, usually one would presume these two methods to be very similar."

We agree which is why we were confused on DREAMzs not being able to converge as well. The calibration was repeated with increased iterations and with multiple datasets, with similar results where the DREAMzs does not converge across the board while DEzs does. We attempted to be honest here on not knowing, but rewrote the sentence to remove speculation on BayesianTools and now it reads starting from line 543:

'We were not able to determine the reason for this in our tests here, specifically was it something related to the behaviour of the parameter space or to some aspect of the technical implementation.'

"420 One could also read your results as showing that you should directly include prior information if you have it, or else you might get a worse / non-sensible result."

Agreed. But to elaborate, the challenge is that the prior information on the system is so complicated that it is possible to get non-sensible parameter values even when using prior data considered reasonable. Because of that we decided to be less constrictive with the priors.

As an example, when extracting different carbon components in a laboratory, N pool is the most difficult to extract, but A is the second most difficult to extract and E is actually the easiest to extract. As a result, based on that lab process the most logical order of recalcitrance is E-W-A-N, which should reflect on the decomposition rates for those pools and thus the prior information would be that E has a higher decomposition rate than A pool. However, our calibration process consistently indicates that A pool carbon compounds are broken down at a much higher rate than E pool compounds. Deeper examination of the field measurement data also supports this as the fraction data in the ED dataset shows that to be true as does very small litterbag datasets that were fractioned into the pools after decomposing for a few years. The current understanding, and we had to dig into this because of another recent manuscript (Currently submitted to Nature Communications), is that the conditions in nature for the decomposition are so different than they are in the laboratory environment that breaking down A pool compounds are actually easier on the field than E pool compounds. Thus in this case if we had applied that stricter prior from the lab information here, we would have produced an unrealistic parameter set as well.

That is why we are explicitly advocating for analyzing the parameter sets afterwards and what they imply for the system behavior as even with the H pool while we did constrain the decomposition rate, it was in truth due to the combined effect of all the parameters affecting the H pool. To stress this in the manuscript, we expanded the sentence here starting from line 562 to:

'All these examples illustrate that prior information and expert opinion should directly inform the calibration and the calibration results themselves should further be reassessed in their physical meaning'

"435 You could of course also think about re-weighting the different data streams. It is not necessary to cite it here, but I think the discussion in Oberpriller et al., 21 could be useful in this section."

In this particular case discussed here, the Hobbie3 dataset is a litterbag experiment similar to the others included in CIDET and LIDET datasets. Consequently, there is no reason to consider that it should be re-weighed so that it has an impact on the global calibration as it is ultimately a question of just the sheer amount of data and joint calibration approach in this case. We did add the re-weighing as an option in the list of other approaches that should be examined. This paragraph now reads from line 513 as follows:

'The inclusion of the Hobbie3 dataset did not meaningfully impact the calibration results (Not shown), which is reasonable considering how small that litterbag dataset (N=192) is compared to the totality of the other datasets (N=~17 000 of which $N_{litterbag}$ = ~12 000) being used in the calibration. This indicates that due to the sheer size of the global calibration data set, smaller local data sets cannot effectively be used just by adding it to the joint calibration process. Additionally, while the smaller datasets such as the Hobbie3 datasets contain insight for their specific locations, they are similar measurements as the ones within CIDET and LIDET and, thus, there is no reason to believe they would provide additional insight to the global application. There are other options, though, by either using the globally estimated parameter ranges as the priors for a calibration with the local data, re-weighing the different datasets based on expert opinion (Oberpriller et al., 2021) or employing a hierarchical calibration approach (Tian et al., 2020, Fer et al., 2021), but the impact of these approaches should be separately researched and tested. Our study still successfully provided a global parameter set that increases the applicability of Yasso model and informs global SOC estimates.contain insight for their specific locations, they are similar measurements than the ones within CIDET and LIDET and, thus, there is no reason to believe they would provide additional insight to the global application.'

---

## Author Response (AR2)

Dear Editor,

Attached is the new version of the manuscript "Calibrating the soil organic carbon model Yasso20 with multiple datasets" with the requested minor revisions. We hope that these revisions will address the remaining issues before the final acceptance for publication.

The responses to the specific requests are below and the lines given are for the revised manuscript.

"Dear editor, dear authors,

I was pleased to find that the revision done by the authors addresses all critical points that were raised in the previous round of reviews. I do not see any further critical issues that would require changes.

Before a possible publication, I would only recommend the following (optional) minor changes:"

Our gratitude once again for the helpful suggestions and for finding the resulting changes to be acceptable. We have also made additions to the manuscript according to the recommendations.

"Abstract: I would make it clear that this is a new Yasso version, so write NEW Yasso 20, and I would mention that you now also compare to the older model version"

We adjusted the abstract based on this and starting from line 10, the abstract now reads:

'**Abstract.** Soil Organic Carbon (SOC) models are important tools for assessing global SOC distributions and how carbon stocks are affected by climate change. Their performances are, however, affected by data and methods used to calibrate them. Here we study how a new version of Yasso SOC model, here named Yasso20, performs if calibrated individually or with multiple datasets and how the chosen calibration method affected the parameter estimation. We also compare Yasso20 to the previous version of the Yasso model. We found that when calibrated with multiple datasets, the model showed a better global performance compared to a single dataset calibration. Furthermore, our results show that more advanced calibration algorithms should be used for SOC models due to multiple local maxima in the likelihood space. The comparison showed that the resulting model performed better with the validation data than the previous version of Yasso.'

"Likelihood: I accept your explanation regarding the fixed sd, and that it doesn't make a large difference, but would still argue that it is fundamentally more appropriate to adjust the sd parameter during the calibration, rather than fixing it to the empirical standard error."

As was the case during the previous revision round, we inherently agree with the reviewer on this. Even though we tested adjusting the sd parameters during calibration and found this to be a very useful exercise, we feel there needs to be a more thorough analysis of the uncertainty calibration as a separate study. For example, CIDET is a much more consistent dataset than LIDET in terms of measurement noise. During this exercise we found that calibrating the uncertainties would reduce the RMSE with LIDET validation dataset, but not with CIDET. We would like to study these interesting findings before we report them as the reviewer accepts our reasoning behind fixing the sd.

Accordingly, we added a part to the Discussion about the need of this kind of research. Starting from line 465, the manuscript now reads:

'Our results show that simultaneously using multiple datasets from different environments improves the general applicability of the SOC model even when the simplistic leaching factor approach had to be used to be able to compare different litter bag datasets and detailed uncertainty estimates were lacking, confirming our first hypothesis. This is in line with prior studies arguing for larger representation in the calibration data (Zhang et al., 2020). Furthermore, a more detailed analysis of different calibrations shows (Figure 2) that the information from multiple datasets is in truth even necessary for the calibration as when calibrating only with one dataset, the decomposition parameter uncertainty ranges either were large or, in the case of the more nuanced EuroDeco dataset, don't even appear to converge. Something that was not examined in this study was how the uncertainties for the different datasets should be defined. Even if the assigned measurement uncertainties were correct for each dataset, combining them introduces structural uncertainties that should also be accounted for (MacBean et al., 2016). A potential method to address would be to estimate the dataset uncertainties along with the model parameters, as done for example in Cailleret et al. (2020) but applying this approach to the SOC system will require a more thorough analysis in order to assess how it impacts the results.'